


# How economically and environmentally viable are multiple dams? A hydro-economic analysis using a landscape-based hydrological model at the basin scale

Anjana Ekka[1,3], Yong Jiang [2], Saket Pande[1], Pieter van der Zaag [1,2]

[1] Department of Water Management, Delft University of Technology, Delft, The Netherlands

[2] IHE Delft Institute for Water Education, Delft, The Netherlands

[3] ICAR-Central Inland Fisheries Research Institute, Barrackpore, India

*Correspondence to*: Anjana Ekka (A.Ekka@tudelft.nl)

**Abstract.** The construction of dams threats the health of watershed ecosystems. Addressing the health challenge requires a clear understanding of the hydrologic effects of multiple dams with concurrent disturbances at the basin scale and their impact on watershed ecosystems. The purpose of the study is to examine the hydrologic, ecological, and economic impacts of multiple dams by analyzing the economic and environmental performance of different combinations of spatially located reservoirs of varying sizes in the Upper Cauvery River basin in India. The approach

uses a previously developed model that integrates a landscape-based hydrological model with an embedded reservoir operations model. Further, the hydrological model is linked to ecological and economic analyses. The combined hydrological impacts of different combinations of reservoirs are quantified using Indicators of Hydrologic Alteration (IHA). Additionally, the production of two major ecosystem services i.e. fish species richness and agricultural production, that depend on flow regimes is estimated, and a production possibility frontier for the two services is

established. Results show that smaller reservoirs on lower-order streams that maximize the economic value of water stored are better for the basin economy and the environment than bigger reservoirs. Growing high-value crops in a command area can maximize the value of stored water and, with lower storage, generate similar economic value while reducing hydrological alterations. The proposed approach can help water and river basin managers to understand the provision of ecosystem services in hydrologically altered basins, optimize dam operations, or even prioritize dam

removal with the balanced provision of ecosystem services.

## 1. Introduction

Population growth, economic development, and climate change have necessitated the construction of water conservation projects such as dams and reservoirs to meet the societal needs for water, food, and energy among others (Suwal et al., 2020; Vanham et al., 2011). A large number of cascade reservoirs, i.e., multiple reservoir dams constructed along a river network, have already been built, and many more are in the process of construction (Suwal et al., 2020). The establishment of reservoirs and dams can alter basin hydrological conditions, particularly river flows

downstream of these dams, by storing and releasing river water that can affect aquatic ecosystems in the basin.

Understanding the impact of multiple dams is important for the sustainable development of river basins. The flow regime of rivers is considered a key factor that is affected by dams while determining river ecosystem health (Richter et al., 1996; Brauman et al., 2007). Many scholars have used the degree of hydrological alteration to measure the

hydrological impact of dams (Gierszewski et al., 2020; Lu et al., 2018, Mittal et al., 2016; Song et al., 2020). While hydrological alterations from dams have basin-wide implications, impact assessment typically concentrates on river segments, assessing the impact upstream or downstream of single dam projects (Nilsson and Berggren, 2000). The assessment becomes more challenging when there are more critical ecosystems affected in the presence of divergent interests of both upstream and downstream stakeholders (Arias et al., 2014).


There are ecological-economic models that analyze tradeoffs between economic development and ecological conservation or among ecosystem services, but they usually consider the effect of a single reservoir (Lu et al., 2015; Rodríguez et al., 2006; Fanaian et al., 2015) or quantify tradeoffs between energy production and environmental degradation (Null, et al., 2020, Song et al., 2019, Wild et al., 2019, Schmitt et al., 2018). Few studies have targeted

multiple dams (Ouyang et al., 2011; Wang et al., 2019; Zhang et al., 2020). For example, Ouyang et al (2011) studied the impact of cascade dams on streamflow, sand concentration, and nutrient pollutant discharge in the upper reaches of the Yellow river. Similarly, Zhang et al., (2020) focused on understanding the hydrologic impact of cascade dams in a small headwater watershed under climate variability. However, there are no studies that assess the impact of multiple dams on the provision of ecosystem services at the basin scale.


This paper evaluates the hydrologic, ecological, and economic impacts of multiple dams by analyzing the economic and environmental performance of different combinations of spatially located reservoirs of varying sizes in the Upper Cauvery River basin in India. The approach developed can be applied to any other river basin in the world. This research is a continuation of the study conducted on the upper Cauvery by Ekka et al., 2022 on individual reservoirs.

In that study, the landscape-based hydrological model was used to analyze primarily the hydrological effects of reservoirs, and the modelled flow regimes with and without reservoirs were then contrasted using hydrological indicators. The current study integrates all four reservoirs and examines the combined hydrologic, ecological, and economic impacts of various combinations of spatially located reservoirs of varying sizes. The motivation is to illustrate the impact of cascade dams on the provision of ecosystem services in basins and to provide an approach to

analyze and optimize dam operations and development that balance the provision of multiple ecosystem services.

The paper is structured as follows. The methodology is discussed in section 2 which includes the integration of reservoirs and the construction of tradeoff between fish species richness and agricultural production. The results are subsequently presented in section 3, and discussed in section 4. The paper concludes with a discussion in section 5 on

possible socio-economic impacts and future implications for dam operation for sustainable production of ecosystem services in the Upper Cauvery River basin.

## 2.   Methods

The main objective of the paper is to evaluate the hydrologic, ecological, and economic impacts of multiple dams in

the study basin. For this objective, a landscape-based hydrological model is formulated and integrated with a reservoir model (Figure 1). Such an integrated model is then used to assess the effect of reservoirs on the flow downstream for varying configurations of considered reservoirs. The hydrological impacts of reservoirs are assessed using the Indicators of Hydrological Alterations. The biophysical quantification of two major ecosystem services, fish species richness, and crop production, supported by the river are estimated, and a production possibility frontier is created to

outline the tradeoffs between the ecosystem services.

### 2.1 Description of the study area

The Cauvery River is the fourth largest river in peninsular India that originates from Talakaveri in the Kodagu district of Karnataka state India. The river has a drainage area of 81,155 km$^2$, which is nearly 2.7% of the total geographical

area of the country (India, WRIS, 2015). The Cauvery basin extends over the Indian states of Karnataka (42%), Kerala (4%), and Tamil Nadu (54%) including the Karaikal region of Puducherry before draining into the Bay of Bengal. Agricultural land is dominant in the basin, with an area of 53700 km$^2$ (or 66 %), which is followed by forest area at 16600 km$^2$ (or 21 %) (Sreelash et al., 2014). Other ecological-economic models analyze land use containing fallow land, scrubland, scrub forests, rivers, streams, and canals, as well as rural and urban mining, swamps, and mangroves.

The Cauvery River has supported irrigated agriculture for centuries and served as the lifeblood of ancient kingdoms and modern cities of South India. Along certain stretches of the Cauvery River, extensive abstraction of water is carried out for intensive agriculture (Vedula, 1985; Bhave et al. 2018). Paddy is the most significant crop in this region, although Ragi, Jawar, and other millets are also grown in rainfed circumstances. More than 60 percent of the total population in the Cauvery basin lives in rural areas with agriculture as the main occupation and 48% of the area under

cultivation (Singh, 2013).

The states of Karnataka, Tamil Nadu, and Kerala, along with the union territory of Puducherry, all claim a share of water from the Cauvery River (Figure 2). Karnataka and Tamil Nadu have competed over the Cauvery River and water distribution throughout the years. The Cauvery dispute has been greatly politicized by both states. The

population depending on the basin increased from 28.6 million in the year 1921 to 50 million in the year 2011 (Down to earth, 2016). Floods and drainage complexity, sewage disposal, sand mining, intensive cropping, salt intrusion of surface and groundwater, and excessive abstraction of groundwater are all challenges that aggravate the water problem in the Cauvery River basin.

Based on the availability of the data needed for the study, the four largest reservoirs in the Upper Cauvery region by gross storage capacity are selected for investigation, including Harangi, Hemavathi, Kabini, and Krishna Raja Sagara (Figure 3). Among the selected reservoirs, Harangi is the smallest reservoir and Krishna Raja Sagara is the largest reservoir in terms of gross storage capacity and command area (Figure 4).




## 2.2 Construction of the hydrological model

Topography is one of the main characteristics of river landscapes that emerge from the coevolution of vegetation and soil with climate (Savenije, 2010; Gao et al., 2014). It determines dominant hydrological processes in a catchment

(Gao et al., 2014), and has been used as a strong constraint in determining and transferring hydrological model structures in space and time (Gao et al., 2016). For these reasons, a landscape-based hydrological model, i.e. Flex-Topo, is used to capture the hydrological process of river landscapes in this study. Each of the sub-basins of selected reservoirs is divided into two parts corresponding to the areas upstream and downstream of the reservoirs, respectively. The landscape of these areas is classified into hillslopes, plateaus, and wetlands, which are further classified based on

their coverage by croplands or forests. Five Hydrological Response Units (HRUs) are then determined, and hydrological model structures specific to these HRUs are formulated to determine 'unaltered' flow regimes (see Ekka et al. (2022) for further details).

## 2.2 Reservoir model

The operation of multi-purpose reservoirs is governed by the objective of meeting the demands of end-users based on certain allocation priorities. Depending on end-user demands, the following conservation of mass equation governs each time step:

$$\frac{S_{t+1} - S_t}{\Delta t} = I_t + O_t - E_t + P_t - (L_t \, X \, D_t) \tag{1}$$


Where $S_t$ =storage, $I_t$=Inflow, $O_t$=outflow, $E_t$ =evaporation on reservoir surface, $P_t$=precipitation on reservoir surface, demand for reservoir water, $L_t$= fraction supply of the demand for the reservoir on day t and $\Delta t$ = 1 day. The reservoir model is embedded in the FLEX-Topo model by using the modeled outflow from the upstream area as an inflow into the reservoir and using the reservoir outflow as an inflow in order to model the runoff at the gauge station generated

out from the downstream area of the reservoir.

The reservoir operation is based on shortage rule curves that define zones within which specified proportions of the demand are covered (Basson et al., 1994). The reservoir operating rules determine $L_t$. $D_t$ is determined based on water

demand calculation for irrigating crops in command areas or for generating hydropower (see Ekka et al. (2022) for

further details).

**2.3 Data requirement**

Rainfall and potential evapotranspiration are used as forcing data. Daily gridded rainfall (0.25° x 0.25°) and

temperature (1° x 1°) data were obtained from the Indian Meteorological Department, Government of India (Pai et al.,

2014; Shrivastava et al., 2009). The rainfall and temperature information are extracted for each sub-basin to force

(run) the FLEX-Topo model and the reservoir model. The potential evapotranspiration ($ET_o$) was calculated based on

the Hargreaves equation (Hargreaves and Samani, 1982) considering max, mean, and min temperature values. The

runoff data were acquired from the Central Water Commission, Government of India. The data on reservoirs including

inflow, outflow, and storage level was accessed from Karnataka State Natural Disaster Monitoring Centre,

Government of Karnataka, India. (https://www.ksndmc.org/Reservoir_Details.aspx ).  Data on acreage and average

productivity of crops at the district level in the study area were accessed from the Directorate of Economics and

Statistics, Government of Karnataka (https://des.karnataka.gov.in/info-2/Agricultural+Statistics+(AGS)/Reports/en).

Similarly, the price information for crops in each district was accessed from  https://agmarknet.gov.in/.


**2.4 Model calibration**

To calibrate and validate the FLEX -Topo models for the sub-basins of the 4 reservoirs, the dataset of topographic

maps, rainfall, and potential evapotranspiration was used. Specifically, the dataset from January 1991 to December

2010 was used for calibration, and the dataset from 2010 to 2016 was used for validation. The reservoir models were

calibrated using the dataset composed of inflow, outflow, storage, rainfall, and potential evapotranspiration, covering

the period from January 2011 to December 2016. Given that topography controls the model structure, the Flex-Topo

models were calibrated based on streamflow observed at the stations downstream of the corresponding reservoirs

(Figure 5). The Elitist Non-Dominated Sorting Genetic (NSGA-II) algorithm was used to calibrate the model

parameters (Deb et al., 2000).


For each parameter set, the modelled run-offs at the stations were compared with the observed runoffs using negative

Nash Sutcliffe Efficiency (-NSE) and Mean Absolute Error (MAE). The MAE value ranges from 0 to ∞, and lower

values demonstrate better model performances. The negative NSE value between -0.5 to -1 demonstrates good model

performance. The NSGA-II algorithm provides a collection of parameters corresponding to the non-dominated model

performance pairs of -NSE and MAE, which are considered as containing the better-performing parameter sets. The

NSE were observed in the range of -0.53 to -0.80 in the calibration phase and -0.50 to -0.65 in the validation phase

for all 4 reservoirs. Similarly, the value of MAE was observed in the range of 0.92 to 1.36 mm/day in the calibration

phase and 0.86 to 2.05 mm/day in the validation phase and are acceptable.

**2.5 Integration of reservoirs**

Figure 5 shows how the models corresponding to the reservoirs are integrated. The outflows from reservoirs Harangi

and Hemavathi flow through the gauge stations of Kudige and M.H. Halli, respectively, and then into the reservoir

Krishna Sagara Raja Reservoir (KRS). Similarly, the outflow from the reservoir Kabini flows through the gauge

station T. Narasipur and then joins the outflow from the reservoir KRS at the gauge station Kollegal.

For each sub-basin, a landscape model with an embedded reservoir can be calibrated and validated without any

modelling uncertainty propagating from models of reservoirs upstream to those downstream. For example, the sub-

basin corresponding to KRS is delineated by the gauging station Kollegal (and hence the flows are modelled at this

station), except those flows generated by contributing areas corresponding to gauge stations Kudige, M.H. Halli and

T. Narasipur. Such calibrated models of flows (without or without respective reservoirs) at the gauge stations

downstream of each of the 4 reservoirs, instead of observed flows, are then used for simulating flow regimes at the

gauging station Kollegal for various possible configurations of reservoirs upstream.

A total of 16 different combinations were generated by removing one or more reservoirs from the schematic graph

presented in Figure 5, and corresponding flows were modelled at the gauge station Kollegal (see Table 1 for an

overview of the different configurations). The modelled flows were then compared with observed flows at the same

station to understand the impacts of reservoirs of varying configuration on the flow regime and, subsequently, on the

production of considered ecosystem services (see Table 1).




Table 1. Comparison of different combinations of reservoirs by storage volume, purpose, sub-basin area and spatial configuration.

| Scenarios | Reservoir combinations | Reservoir characteristics | | |
|---|---|---|---|---|
| | | Storage volume ($10^6$ m³) | Purpose of the reservoir & Net Command Area (NCA)-ha | Spatial configuration |
| **Scenario with four reservoirs (Base scenario)** | | | | |
| $S_{abcd}$ | A+B+C+D | A: 240.69<br>B: 1050<br>C: 552.74<br>D: 1400.31 | Irrigation - A, B, D<br>Irrigation & Hydropower-C<br>For individual reservoir<br>NCA : 499215 | A, B: upstream & on a tributary<br>C: downstream & on a tributary<br>D: downstream & on main channel |
| **Scenario with three reservoirs** | | | | |
| $S_{bcd}$ | B+C+D | B: 1050<br>C: 552.74<br>D: 1400.31 | Irrigation - B, D<br>Irrigation & Hydropower-C<br>NCA: 445677 | B: upstream & on a tributary<br>C: downstream & on a tributary<br>D: downstream & on main channel |
| $S_{abd}$ | A+B+D | A: 240.69<br>B: 1050<br>D: 1400.31 | Irrigation - A, B, D<br>NCA: 453485 | A, B: upstream & on a tributary<br>D: downstream & on main channel |
| $S_{acd}$ | A+C+D | A: 240.69<br>C: 552.74<br>D: 1400.31 | Irrigation - A, D<br>Irrigation & Hydropower-C<br>NCA: 207350 | A: upstream & on a tributary<br>C: downstream & on a tributary<br>D: downstream & on main channel |
| $S_{abc}$ | A+B+C | A: 240.69<br>B: 1050<br>C: 552.74 | Irrigation - A, B<br>Irrigation & Hydropower-C<br>NCA: 391133 | A, B: upstream & on a tributary<br>C: downstream & on a tributary |
| **Scenario with two reservoirs** | | | | |
| $S_{bd}$ | B+D | B: 1050<br>D: 1400.31 | Irrigation - B, D<br>NCA:  399947 | B: upstream &  on a tributary<br>D: downstream & on main channel |
| $S_{cd}$ | C+D | C: 552.74<br>D: 1400.31 | Irrigation - D<br>Irrigation & Hydropower-C<br>NCA: | C: downstream & on a tributary<br>D: downstream & on main channel |
| $S_{ad}$ | A+D | A: 240.69<br>D: 1400.31 | Irrigation - A, D<br>NCA: 161620 | A: upstream & on a tributary<br>D: downstream & on main channel |
| $S_{cb}$ | C+B | C: 552.74<br>B: 1050 | Irrigation - B<br>Irrigation & Hydropower-C<br>NCA: 153812 | C: downstream & on a tributary<br>B: upstream & on a tributary |
| $S_{ab}$ | A+B | A: 240.69<br>B: 1050 | Irrigation - A, B<br>NCA: 345403 | A, B: upstream & on a tributary |
| $S_{ac}$ | A+C | A: 240.69<br>C: 552.74 | Irrigation - A<br>Irrigation & Hydropower-C<br>NCA: 99268 | A: upstream & on a tributary<br>C: downstream & on a tributary |
| **Scenario with one reservoir** | | | | |
| $S_d$ | D | D: 1400.31 | Irrigation - D<br>NCA: 108082 | D: downstream & on the main channel |
| $S_b$ | B | B: 1050.00 | Irrigation - B<br>NCA: 291865 | B: upstream & on a tributary |
| $S_c$ | C | C: 552.74 | Irrigation & Hydropower-C<br>NCA: 45730 | C: downstream & on a tributary |
| $S_a$ | A | A: 240.69 | Irrigation - A<br>NCA: 53538 | A: upstream & on a tributary |
| **Scenario with no reservoir** | | | | |
| $S_0$ | NO | -- | -- | -- |


**2.6 Indicators of hydrological alterations**

The set of Indicators of Hydrological Alteration (IHA) initially proposed by Richter et al. (1996) is used to measure
the effects of different reservoir combinations on the flow regime in the Upper Cauvery basin. The parameters
considered in IHA have strong relationships with river ecosystems, and therefore can be used to assess the impact of
dams, barrages and other types of water diversion on the flow regime. The IHA are classified into five groups based
on magnitude of monthly flows, magnitude and duration of annual extreme flow conditions, and frequency and
duration of high and low flow rates.

**2.7 Construction of Production Possibility Frontier**

The production possibility frontier (PPF) is an economic concept typically used to express different quantitative
combinations of commodities that can be efficiently produced by an economy under limited production resources
(Martinez-Harms et al., 2015). It can be described as the outward boundary of the convex hull of the production set
of the economy.

In the Upper Cauvery, flows are increasingly being diverted for irrigation purposes, thus leaving less water for aquatic
ecosystems. The tradeoff between irrigation agriculture and aquatic ecosystems can be measured by the production
possibility frontier if indicators can available capturing irrigation agriculture and ecosystems. In this study, the value
of crop production is used to measure irrigation output, and the fish species richness is used to measure the service of
aquatic ecosystems sustained by the flow of the river. Different reservoir combinations correspond to different
partitioning of flows for irrigation and, hence, for aquatic ecosystems. Therefore, different pairs of crop production
value and fish species richness can be generated for different reservoir combinations.

A production set is determined based on possible production outputs of different reservoir combinations. Specifically,
it is defined by the convex-hull of the 16 pairs of the indicators of irrigated production value and fish species richness.
The production possibility frontier is then the outward boundary of the production set.





*2.7.1 Agricultural production*

For agricultural production, the relationship between crop yield and water depends on the corresponding relative reduction in evapotranspiration (ET). The actual yield is calculated based on the following formula by FAO (2012)

$$1 - \frac{Y_a}{Y_o} = K_y \left(1 - \frac{ET_a}{ET_p}\right) \tag{2}$$


Where $Y_a$ = actual Yield, $Y_o$ = optimum Yield, $ET_a$ = Actual Evapotranspiration, $ET_p$ = Potential Evapotranspiration, and $K_y$ = yield response parameter.

Total agricultural production is equal to agricultural output from both rainfed and irrigated areas, with irrigated areas depending on water withdrawn for irrigation. As a result, total agricultural production is a reliable predictor of the amount of water utilized for irrigation. The crop specific prices are multiplied by the corresponding production level to indicate the output or value of the ecosystem service supported by the river.

*2.7.2 Fish Species Richness*

The regime of river flows is an important determinant of the abundance and biodiversity of riparian species, particularly fish species. As river discharge reflects the characteristics of flow regime, species-discharge models are usually used to quantify the impact on species richness of anthropogenic modification of rivers via flow regime

(Xenopoulos and Lodge, 2006). However, the flow regime of a river is composed of several ecologically relevant flow characteristics such as magnitude, frequency, duration, timing, and rate of change of flow events. In other words, flow characteristics other than mean river discharge also play a vital role in sustaining aquatic ecosystems. In this study, we adopted an empirical function below that was identified by Iwasaki et al. (2012) to quantify fish species richness in relation to river discharge. In this method, basin characteristics such as area and latitude are used to predict fish

species richness. The flow characteristics such as coefficient of variation of mean frequency of low flow in a year, coefficient of variation in the Julian date of annual minimum flow and maximum proportion of the year in which floods have occurred are also used. Here floods are defined as events when flows are greater than or equal to flows with a 60% exceedance probability (Olden and Poff, 2003).


Fish species richness $FSR = \exp(3.95 - 0.0342\,LAT + 0.273Area + 0.373MAD - 1.57FL2 + 0.832TH3 -$

$0.116TL2)$ (3)

where LAT = Absolute value of the latitude of the gauge station where flow is measured

Area = $\log_{10}$ transformed basin area (km$^2$)

MAD = $\log_{10}$ transformed mean annual discharge (m$^3$s$^{-1}$)

FL2 = Coefficient of variation of mean frequency of low flow per year

TH3 = Maximum proportion of the year (number of days /365) during which floods have occurred

TL2 = Coefficient of variation in the Julian date of the annual minimum flow.

## 3. Results

This section first reports on the quality of the model developed for the study area. The developed model is then used

to simulate flow regimes for the 16 scenarios of different combinations of reservoirs as shown in Table 1, with the

degree of hydrological alterations assessed. The production of considered ecosystem services is then quantified, and

a production possibility frontier for the considered ecosystem services is derived and discussed. Table 2 provides a

summary of these results.


### 3.1 Impact on flow regimes generated by different combinations of reservoirs

The flow regimes generated by different combinations of reservoirs are disentangled by major hydrological indicators

such as mean annual flow and annual extreme flow conditions and are further classified by storage volume and

reservoir spatial configurations.


### 3.1.1 Flow regimes characterized by storage volume under different scenarios

The highest mean annual flow was estimated for $S_0$ (1,548 m$^3$s$^{-1}$) with no reservoir, followed by $S_c$ (1,460 m$^3$s$^{-1}$) and

$S_b$ (1,377 m$^3$s$^{-1}$) containing only one reservoir in the scenarios (Figure 6). In terms of storage volume, KRS (D) is the

biggest reservoir followed by Hemavathi reservoir (B) and Kabini reservoir (C). KRS (D) in combination with one

another reservoir ($S_{bd}$, $S_{cd}$, $S_{ad}$) and two other reservoirs ($S_{bcd}$, $S_{abd}$, $S_{acd}$) yielded mean annual flows of less than 500

m$^3$s$^{-1}$.

The magnitude of annual extreme conditions, the 1-3-7-30 day minimum and base flow indices were greatly affected

by the construction of reservoirs having bigger storage volumes (Figure 7). However, in scenarios with three

reservoirs, $S_{abd}$ has less impact compared to $S_{acd}$ despite Kabini (C) having less storage capacity compared to the

Hemavathi reservoir (B).

The extreme low peak flow for scenario $S_D$ appears to be the lowest of the scenarios with only one reservoir (Table 2) as KRS (D) reservoir has the largest storage capacity. Similarly, the KRS (D) generated flows with lowest values of extreme low peak conditions in combinations with two ($S_{bcd}$, $S_{abd}$) and three ($S_{abcd}$) reservoirs. However, in combinations with one and no reservoir, despite having varying storage capacities, the extreme low peaks of flows

generated by $S_a$ , $S_b$, $S_{ac}$, and $S_{bc}$ appear to be similar (Table 2).

### 3.1.2   Flow regimes characterised by the use of reservoirs

Kabini (C) is the only reservoir used for hydropower. Scenario $S_c$ generates a mean annual flow that is the second highest, after that of $S_0$ with no reservoir in the basin (Figure 7). The mean annual flows of combined irrigation and

hydropower reservoirs ($S_{ac}$ and $S_{bc}$) are observed to be higher (1,076-1,289 $m^3s^{-1}$) when compared with that of two irrigation reservoirs ($S_{ab}$). Similarly, the mean annual flow of scenario $S_{abc}$ with 3 reservoirs is around 906 $m^3s^{-1}$, which is more than those of the scenarios $S_{bd}$, $S_{cd}$, $S_{ad}$ but less than those of $S_{bc}$, $S_{ab}$ and $S_{ac}$ with two reservoirs. This is because Kabini (C) is a hydropower reservoir, which releases water frequently and ensures flows above a certain threshold resulting in a higher mean.


The comparison of a scenario with two irrigation reservoirs and one hydropower reservoir ($S_{abc}$) to a scenario with two irrigation reservoirs ($S_{bd}$) indicates that the former has less impact on mean annual extreme flow conditions such as 1, 2 and 7-day minimum than the latter. Comparing similar combinations of two reservoirs only for irrigation ($S_{ad}$ and $S_{bd}$) versus those that contain the hydropower reservoir ($S_{cd}$) indicates that the hydropower reservoir decreases the

low pulse count and low pulse duration compared to irrigation reservoirs.



Table 2. Overview of hydrological impact, environmental flow parameters, fish species richness, and value of agricultural production of different scenarios


| Scenarios | Hydrological impact | | | | | Environmental flow Parameters (m³/s) | | Fish species richness (in number) | Economic value of agricultural production (Million Lakh ₹ per year) |
|---|---|---|---|---|---|---|---|---|---|
| | Mean annual flow (m³/s) | Low pulse count (days) | High pulse count (days) | Low pulse duration (days) | High pulse duration (days) | Extreme low peak | Extreme low frequency | | |
| Scenario with four reservoirs Integrated | | | | | | | | | |
| $S_{abcd}$ | 265 | 2.2 | 3.4 | 52.5 | -16.6 | 44.9 | 1.0 | 31 | 1.19 |
| Scenario with three reservoirs integrated | | | | | | | | | |
| $S_{bcd}$ | 296 | 2.4 | 3.6 | 44.1 | -73.3 | 44.9 | 0.9 | 34 | 1.16 |
| $S_{abd}$ | 443 | 1.4 | 3.9 | 90.5 | -16.8 | 66.9 | 0.9 | 35 | 1.16 |
| $S_{acd}$ | 274 | 2.6 | 3.6 | 46.3 | -29.1 | 44.9 | 1.0 | 35 | 1.18 |
| $S_{abc}$ | 907 | 2.3 | 3.8 | 57.7 | -17.1 | 117.0 | 1.4 | 56 | 1.12 |
| Scenario with two reservoirs integrated | | | | | | | | | |
| $S_{bd}$ | 480 | 1.8 | 3.9 | 75.6 | -88.3 | 61.0 | 0.6 | 39 | 1.13 |
| $S_{cd}$ | 310 | 2.2 | 3.5 | 55.9 | -79.7 | 48.7 | 1.1 | 35 | 0.87 |
| $S_{ad}$ | 452 | 1.4 | 4.2 | 90.2 | -89.8 | 67.1 | 1.0 | 43 | 0.87 |
| $S_{bc}$ | 1289 | 2.6 | 3.8 | 46.4 | -29.4 | 181.1 | 1.6 | 61 | 1.05 |
| $S_{ab}$ | 995 | 1.6 | 3.1 | 86.9 | -17.1 | 119.1 | 1.4 | 60 | 1.10 |
| $S_{ac}$ | 1076 | 2.9 | 3.6 | 46.8 | -29.9 | 181.0 | 1.7 | 62 | 0.84 |
| Scenario with one reservoir Integrated | | | | | | | | | |
| $S_d$ | 488 | 1.9 | 4.0 | 74.2 | -91.9 | 60.8 | 0.6 | 54 | 0.85 |
| $S_b$ | 1377 | 2.6 | 3.6 | 42.5 | -103.7 | 181.9 | 1.6 | 63 | 1.08 |
| $S_c$ | 1460 | 2.4 | 3.5 | 42.0 | -95.7 | 242.9 | 2.1 | 80 | 0.82 |
| $S_a$ | 1164 | 2.6 | 3.4 | 48.0 | -29.9 | 182.8 | 1.4 | 79 | 0.82 |
| Scenario with no reservoir | | | | | | | | | |
| $S_0$ | 1548 | 2.4 | 4.0 | 45.0 | -109.7 | 242.9 | 2.1 | 124 | 0.80 |

### 3.1.3 Flow regimes characterised by spatial configuration of reservoirs in combinations

Harangi (A) and Hemavathi (B) reservoirs are located in the upstream areas of the basin, on one of the tributaries of the Upper Cauvery. The arrangement of multiple dams on river tributaries across the basin has an impact on the river flow regime as well. Harangi (A) reservoir is the smallest in terms of volume, followed by Kabini (C), Hemavathi (B), and KRS (D). When comparing the flow altered by combinations with one reservoir, $S_a$ produces regimes with lower mean annual flows than $S_b$. Harangi (A) reservoir has a residence time of 7.2 months. This means that Harangi (A)

reservoir does not hold water for an extended period. Hemavathi (B) reservoir, on the other hand, has a relatively long residence time (22.6 months), Furthermore, in the absence of a reservoir, the mean annual flow in M.H.Halli sub-basin (having Hemavathi (B) reservoir) is lower (75 m$^3$ s$^{-1}$) when compared to Kudige (having Harangi (A) reservoir, 139 m$^3$ s$^{-1}$), T. Narasipur (having Kabini (C) reservoir, 349 m$^3$ s$^{-1}$) and Kollegal sub-basins (having KRS (D) reservoir, 630 m$^3$ s$^{-1}$). M.H.Halli sub-basin's flow regime made little contribution to the overall flow regime. As a result, the $S_a$

scenario generates a lower mean annual flow than the $S_b$ scenario. Similarly, for two reservoir combinations, the M.H.Halli sub-basin has a lower no-reservoir flow regime than the Kudige sub-basin, which contributes less to the overall flow regime. As a result, $S_{ac}$ performs worse than $S_{cb.}$ Among three reservoir scenarios, the mean annual flow and other hydrological parameters of the $S_{bcd}$ and $S_{acd}$ scenarios were equally bad as the four-reservoir scenario.

### 3.2 Agricultural production

The agricultural production in the sub-basins is calculated based on an assumption that irrigated area becomes an unirrigated (i.e., rainfed) one when the corresponding reservoir is removed in a corresponding scenario, without changing the crops that are being cultivated. The different proportion of cultivated and irrigated land is given in figure

340 8.

The district-wise information on crops, areas and prices were taken to estimate the quantity and economic value of agricultural production. The average acreage under each crop and maximum production of crops under irrigated condition in each of the districts were taken to estimate the production values. Then the modelled actual

evapotranspiration values for each sub-basin were used to calculate reduction in yields in rainfed condition of relevant

crops, which grouped into five categories, namely, cereals, pulses, oilseeds, horticultural & plantation (H&P) crops, and spices. Among Horticultural & Plantation crops, the crops like Coffee, Coconut and Cashewnut contribute to 65 percent of the total H&P crops cultivated area. According to current estimates, the contribution of plantation crops accounts for 58 percent of the economic value of the H&P crops.


In terms of economic value, when comparing the individual sub-basin, the horticultural crops and spices contributed more to the economic value of all the sub-basins (Figure 9). In M.H. Halli and Kollegal sub-basins, though the area under cereals is high, the economic value of cereal production is low compared to the horticultural crops and spices. When comparing the economic value based on reservoir and non-reservoir area, not much difference was observed in

the economic value of crop group of pulses, oilseeds, and fibres in all the sub-basins. The difference in economic value with and without reservoir is seen among horticultural crops and spices in three sub-basins including Kudige, M.H. Halli and T. Narasipur sub-basins. In Kollegal sub-basin, the cultivated area accounts for 47 percent of the total sub-basin and out of which only 10 percent accounts for irrigated area. therefore, not much difference is seen in the economic value of scenario with and without a reservoir.


Figure 10 indicates that the economic value from agricultural production varies across scenarios of dam development. In general, increasing the number of dams does raise the economic value of agricultural production as compared to scenario $S_0$ without a dam. The scenario of 4 dams generates the highest economic value from agricultural production. However, the value contribution of alternative dam planning and design scenarios differs. For example, the scenario

of 4 dams does not show a dramatic value increase as compared to the scenarios of 3 dams. Among the scenarios of 2 dams, there are 3 scenarios, i.e $S_{bd}$, $S_{bc}$, and $S_{ab}$, that show much higher-value generation than other scenarios and that are comparable to scenarios of 3 and 4 dams. In the case of 1 dam, scenario Sb shows a much higher economic value generation. This is because the Hemavathi reservoir (B) has a well-developed command area growing mainly horticultural crops that fetch high prices in the market thereby increasing the economic value.


### 3.3  Fish species richness across sub-basins

In addition, to mean annual flow, flow characteristics such as magnitude, frequency, duration, and timing, play a vital role in sustaining aquatic ecosystems. Therefore, the value of fish species richness (FSR) is quantified based on river flow characteristics (FSR-IHA) for all the combinations.


The results of species richness value for different combinations of the reservoirs are shown in Figure 11. The FSR-IHA method shows larger variation in FSR values for different combinations. The FSR is greatly impacted by the combinations that contained a large reservoir (such as KRS) due to significant decrease in mean annual flow and in the coefficient of variations of low flow frequencies. This can be seen in the combinations containing 1 ($S_d$), 2 ($S_{bd}$,

$S_{cd}$, $S_{ad}$) and 3 ($S_{bcd}$, $S_{abd}$, $S_{acd}$) reservoirs where lower FSR is observed. Among the scenarios of two reservoirs, $S_{ad}$ has better FSR than $S_{bd}$ despite having lower mean annual discharge demonstrating the effect of other hydrological flow regime parameters on FSR. Among the combinations containing 3 reservoirs, not much difference in FSR value is observed except Sabc which scores higher than others ($S_{bcd}$, $S_{sbd}$ and $S_{acd}$). These combinations contain KRS, which is the most downstream and the largest reservoir and contains two smaller reservoirs out of three in various spatial

configurations upstream of the KRS reservoir. This shows that a very large reservoir can dominate the effect of reservoirs on the flow regime characteristics and consequently on envrionmental flows.

### 3.4 The production possibility frontier (PPF)

The production possibility frontier (PPF) between agricultural production and fish species richness for different combinations of the reservoirs is shown in Figure 12. As can be seen from the figure, FSR - agricultural production combinations are used to define the convex hull of the production set. The PPF is then defined as the outward boundary of the production set.

The findings show that the scenario without any reservoir ($S_0$) is advantageous for the diversity of fish species. Due to lower value from agricultural production, scenarios with 1 reservoir ($S_d$, $S_a$ and $S_c$) and 2 reservoirs ($S_{cd}$, $S_{ad}$, and $S_{ac}$) do poorly with respect to the frontier. However, due to lower value in fish species richness, scenarios with four reservoirs ($S_{abcd}$), three reservoirs ($S_{bcd}$, $S_{abd}$, $S_{acd}$) and two reservoirs ($S_{bd}$ and $S_{bc}$) are also considered inferior with respect to the frontier. The scenario $S_{bc}$ is however slightly worse off in terms of species richness and agricultural

production, relative to the PPF curve.



Five scenarios $S_0$, $S_b$, $S_{ab}$, $S_{abc}$, $S_{acd}$, and $S_{abcd}$ define the frontier. The scenario with all reservoirs ($S_{abcd}$) produces the highest agricultural output but has the least diversity of fish species. The scenario Sb is the only one with a single reservoir (Hemavathi reservoir B) that has the highest area under irrigation with high-value crops. The scenarios Sb,

Sab, and Sabc do not include the KRS (D) reservoir with a maximum storage volume, and thus the flow regime was not significantly altered as compared to the cases of $S_{abcd}$ and $S_{acd}$. This resulted in better diversity of fish species and a better 'balance' between agricultural production and FSR. The scenario $S_b$ features 1 reservoir with the largest command area producing high-value crops when compared to scenarios $S_{ab}$ and $S_{abc}$. Furthermore, both $S_{abc}$ and $S_{acd}$ are on the frontier because the KRS (D) reservoir in the scenario $S_{acd}$ adversely altered the flow regime by diverting

more water for agriculture, thereby boosting agricultural production but simultaneously limiting the diversity of fish species.

## 4. Discussion

### 4.1 Hydrological impacts of reservoirs on flow regime

The analysis of IHA for different combinations of reservoirs revealed that the storage volumes of reservoirs impact the mean annual flow. For example, when reservoirs are combined with KRS (D), which has the highest storage volume, an enormous decline in mean annual flow is observed. The findings are consistent with the study conducted in the Lancang river mainstream in China where dams with storage capacities greater than 100 million m$^3$ had stronger impacts on streamflow regimes than smaller ones (Han et al., 2020).


In contrast, hydropower dams appear to have less impact on low flow pulses compared to irrigation reservoirs. Previous studies indicated that hydropower dams cause monthly mean water levels to rise during the dry season and fall during the wet season (e.g., Hecht et al., 2019). Even though the dry and rainy years were not compared in the current study, the monthly mean annual flow hydropower reservoirs were estimated to be higher throughout the year.

In scenarios of two and three reservoirs, combining irrigation reservoirs with a hydropower dam has less impact on river flow regimes than combining reservoirs that only serve for irrigation purposes. One obvious explanation is that hydroelectric dams frequently release water to produce energy, maintaining river flows year-round above a certain threshold.

The flow regime of a tributary can compensate for a reservoir's low flow in the tributary and result in a lower impact on the overall flow regime. In the present study, Harangi river flows compensated for the flow altered by Hemavati reservoir in the resulting flow regime downstream of the reservoir.  A similar study in the Eden Watershed in the U.K. found that tributaries significantly contributed in controlling flooding in downstream areas of the watershed (Pattison et al., 2014).


### 4.2 Social and ecological impacts

Change in flow regime has a direct impact on fish species richness (FSR). Since mean annual flows directly affect FSR, the scenarios containing the largest reservoir (KRS-D) had significant negative impacts on FSR due to declines in mean annual flows and the coefficient of variation of the low flow frequency. When comparing scenarios that

contained the hydropower reservoir with scenarios containing only irrigation reservoirs, the FSR values were higher
in the former indicating that irrigation reservoirs more adversely alter the flow regime. Further in contrast to two
reservoir combinations, there was a significant difference in the FSR values amongst the scenarios containing three
reservoirs due to a greater alteration in flow characteristics resulting from the three reservoir combinations.

In contrast, no significant difference in the economic value of agricultural production for different scenarios were
observed based on storage volumes, the purpose of the reservoirs, and the orders of the streams on which the reservoirs
are constructed. The economic value of agricultural production appears to be largely influenced by the area irrigated
per unit volume of stored water in the reservoir. This means that if water is being stored then it should be used as
efficiently as possible, i.e. by producing high value agricultural products, to maximize its value.


In the present study, the average contribution of a reservoir to agriculture production was estimated to be ₹ 0.40 billion
per year (*$ 0.005 billion per year). It not only supports food security but also contributes to economic development
and growth. Most of the horticultural crops and spices that are grown in the Upper Cauvery basin are exported outside
the country and earn foreign exchange. Fishing is another important ecosystem service supported by the river flow.

The economic value of both commercial and subsistence fishing of the Cauvery River is estimated to be ₹35.93 billion
per year ($ 0.44 billion per year) (Pownkumar et al., 2022). Nearly 4,395 active fishermen directly depend on the
Cauvery River for their livelihood (Pownkumar et al., 2022). Ecologically, fisheries also play an important role in
regulating the food web dynamics of the riverine and floodplain ecosystem (Pownkumar et al., 2022). For example,
in freshwater systems, the feeding behaviour of many adult and young fishes has cascading effects on population

dynamics down the food web (Carpenter et al., 1985). They play an important role in the structure and functioning of
the microbial food web (Felip et al 1996; Simon and Wunsch 1998). Although fish populations have a significant role
in sustaining the river environment, their direct economic contribution to human wellbeing is lower than that of
agriculture. But the ecological importance of fisheries in maintaining ecosystem services and functioning, which is
indirectly supported by fish species richness, is often ignored in river basin management decisions. The presented

tradeoff approach of evaluating the PPF between FSR and agriculture production provides a means to consider both
in a balanced manner.

    *1 dollar is equal to 81.66 rupees as on 6 October, 2022

### 4.3 The role of PPFs in decision making

The production set in Figure 12 shows the different combinations of two ecosystem services that can be produced using available water resources. Agricultural production and fish species richness are considered as functions of river flow characteristics that depend on how flows are partitioned into irrigation and environmental flows by the reservoirs. There are combinations that lie on the production possibility frontier and represent efficient production of ecosystem services. All these combinations included Hemavathi (B) reservoir, which has the largest command area growing high-value crops except Sabcd that has the highest volume stored. This optimally benefitted the production of both services, making it Pareto-superior to other combinations.

The analysis revealed that large dams that do not maximize the value of water stored, i.e. by growing low-value crops in smaller command areas, affect both FSR and the economic value of agricultural production adversely. Such reservoirs are least favorable, as they are strictly Pareto inferior to other combinations. In contrast, smaller reservoirs on tributaries (away from the main river stem) that grow high-value crops and maximize the value of water stored are Pareto superior and most preferred. Small reservoirs would significantly increase the value of the water while having a minimal detrimental effect on areas upstream and downstream (Van der Zaag and Gupta, 2008). For decision-making, this means that large reservoirs that do not maximize the value of water stored should be discouraged and smaller more effective reservoirs should be encouraged if faced with a choice between the two types of reservoirs. However, larger reservoirs are substantially less expensive (per $m^3$) than smaller reservoirs due to economies of scale, and as a result, the ecological costs must be included during the cost-benefit analysis. (Van der Zaag and Gupta, 2008). Additionally, in the study, an assumption was made that an irrigated area becomes unirrigated (i.e., rainfed) when the associated reservoir is withdrawn creating different scenarios. This assumption could have impacted the economic value of different scenarios to some extent as farmers may change production practices in response to the unirrigated condition.

### 4.4 Ecosystem service perspective on PPF and future challenges

Understanding ecosystem service (ES) interactions was achieved through the interpretation of the production possibility frontier. However, the complexity of the interactions may prevent the translation of ES knowledge into

decision-making processes (Vallet et al., 2018; Hegwood et al., 2022). In the present study, the scenario without

reservoirs ($S_0$) was hydrologically a superior choice in terms of fish species richness. However, it had the lowest

agricultural output, which would negatively affect employment generation and economic growth. Similar to this, the

integration of all four reservoirs ($S_{abcd}$) would boost agriculture production by increasing the area of land irrigated but

at the expense of lower fish species richness that would be detrimental to riverine ecology. The combination $S_b$, $S_{ab}$,

and $S_{abc}$, which can enhance both ecosystem services, yield more balanced results.

However, intangible services were not analysed in this study. For example, humans directly consume or use both

agriculture and fisheries products for food, nourishment, and employment, and to support their way of living. Both

agroecosystems and fisheries provide regulating and supporting services that are crucial for ecosystem functioning

and resilience. However, the human-driven ecosystem dis-service from agricultural activities can reduce ecosystem

resilience and decrease service generation that are necessary for human survival. Therefore, the non-tangible ES and

dis-services should also be taken into consideration using appropriate economic valuation tools in a tradeoff analysis.

Further, there is a need to determine which efficient ES combinations would be preferred by stakeholders by assessing

indifference curves that describe human preferences for different ecosystem services including regulating and

supporting services (Cavender-Bares et al., 2015; King, et al., 2015).

## 5   Conclusion

The main objective of the paper was to evaluate the hydrologic, ecological, and economic impacts of multiple dams

in the Upper Cauvery River basin. To do so, an approach was formulated to estimate the production of river ecosystem

services using a landscape based hydrological model with the modelling of reservoir operations embedded in it. Such

an integrated model was then used to assess the effect that reservoirs have on the flow by considering various scenarios

of the reservoirs in a virtual experiment.

The hydrological impacts of different combinations of reservoirs were assessed using Indicators of Hydrological

Alterations. The biophysical quantification of major ecosystem services, fishes and crop production, supported by the

river were estimated and a production possibility frontier was quantified to understand the tradeoff between the


ecosystem services. The main findings that can enhance our understanding of the effect of multiple dams on the provision of ecosystem services and help optimize river management plans are summarized below.

• The mean annual flow and annual extreme conditions of minimum and maximum flows are adversely affected by the largest storage dam. In comparison to reservoirs used just for irrigation, scenarios of reservoirs used for hydropower and irrigation have less impact on low-flow pulses and low-flow duration.

      • Large dams that do not maximize the value of water stored, i.e., by growing low-value crops in smaller command areas, affect both FSR and economic value from agricultural production adversely. Such reservoirs are the least

530       favourable and should be discouraged by policy makers.

      • Growing high-value crops with a highly established command area of small and medium reservoirs can strike a favorable balance between agricultural production and fish species diversity.

      • Heavily altering the river landscape with reservoirs (e.g., by having all the reservoirs) provides a superior result in the sense that it generates the most possible agricultural income. However, it may not be preferred by diverse

535       stakeholders such as fishers and environmentalists due to dismal environmental flows. Such an option produces the lowest FSR and degrades the environment the most and perhaps should be favoured less than a combination of reservoirs that strikes a favorable balance between agricultural production and fish species diversity while still efficiently producing both.

**Competing interests**:  One of the co-authors is a member of the editorial board of the journal Hydrology and Earth System Sciences. The peer-review process was guided by an independent editor, and the authors have also no other competing interests to declare.

**Acknowledgements**

This research was financed by the Indian Council of Agricultural Research, Ministry of Agriculture Government of India through a scholarship for the PhD study [18(26)/2016-EQR/Edn] for the principal author. The authors are thankful to National Data Centre, Central Water Commission, New Delhi for providing gauge station data and GIS-related information on the Cauvery River basin. The authors wish to acknowledge the help provided in this research by the water efficiency task force officers under the India-European Union Water Partnership (IEWP).



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






**Figure 1. Overview of the methodologic structure of the study**





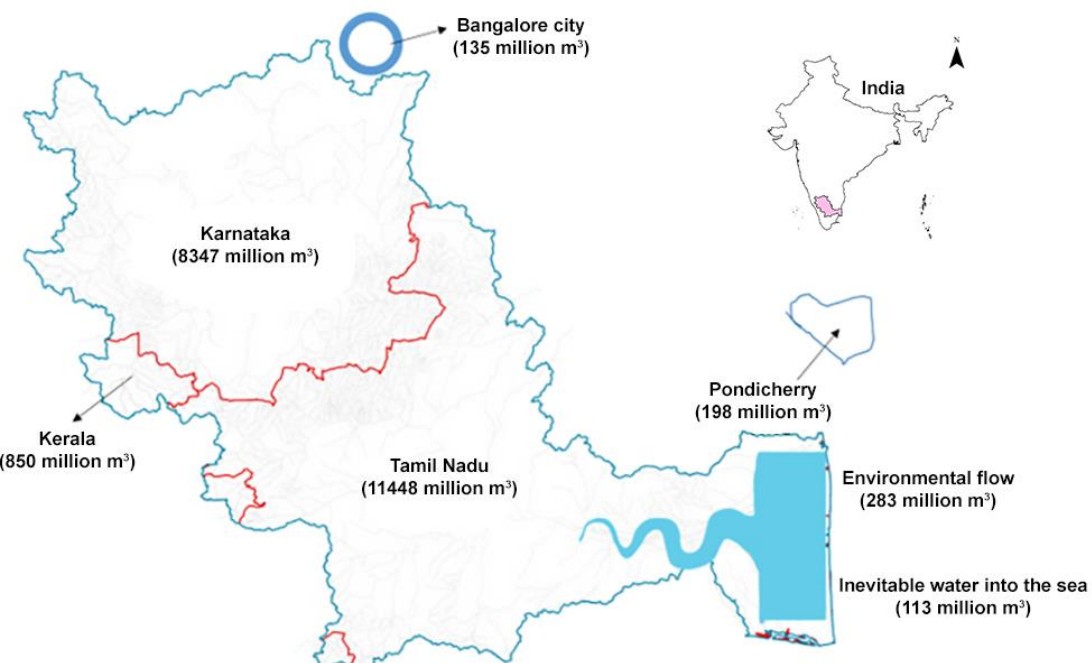

**Figure 2.** Overview of water allocation (million m³/year) in the Cauvery basin among different states/union territories as per the supreme court Verdict in 2018.


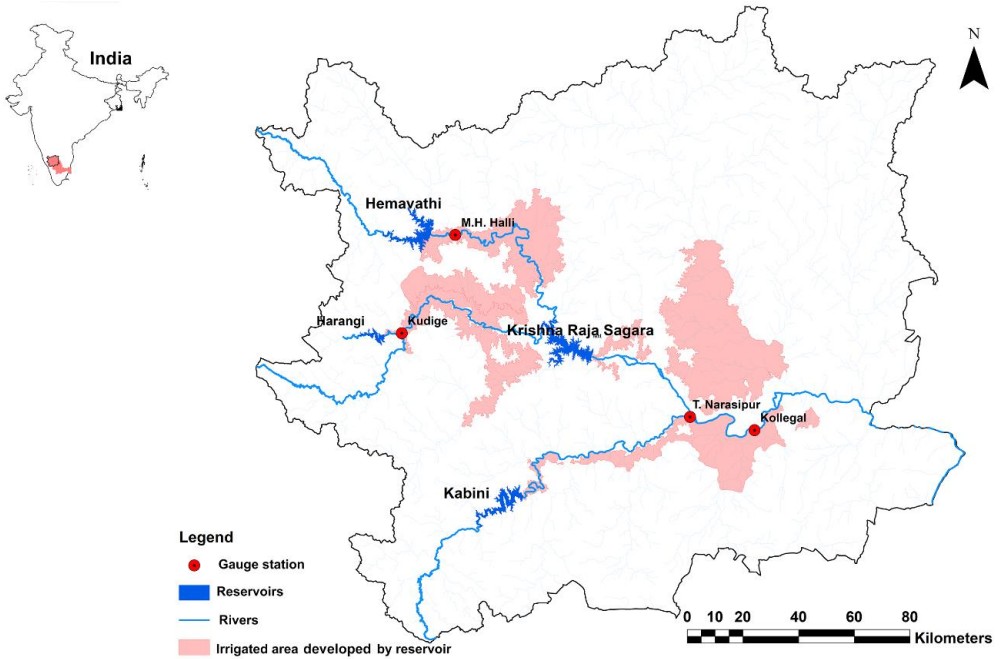

**Figure 3. Overview of reservoirs with sub-basins delineated based on gauge stations in the Upper Cauvery basin. Gauge stations are located downstream of the reservoirs.**






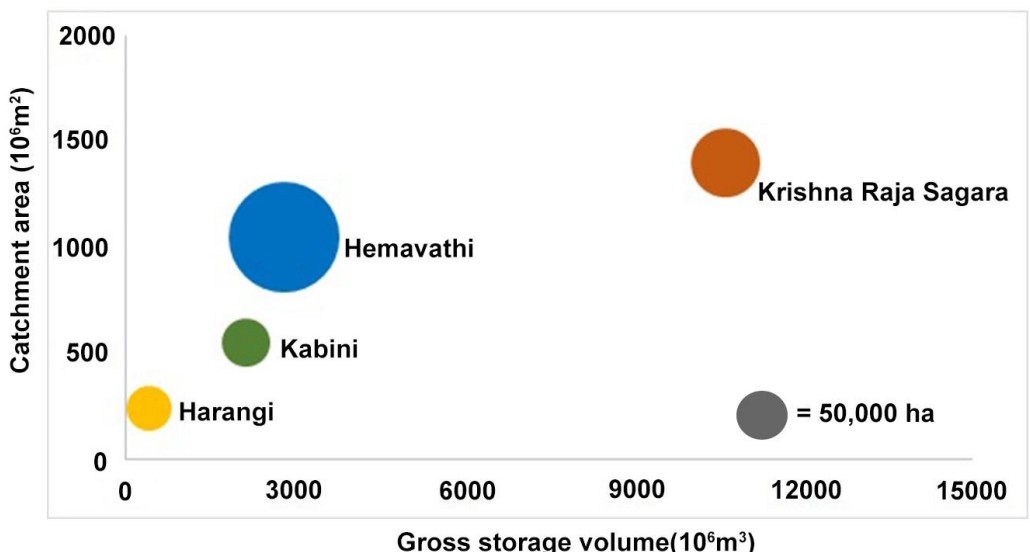

**Figure 4. Overview of selected reservoirs by catchment area and gross storage volume. The size of the bubbles is proportional to the size of the catchment areas. The grey circle indicates the size of the bubbles which is equivalent to 50,000 ha.**





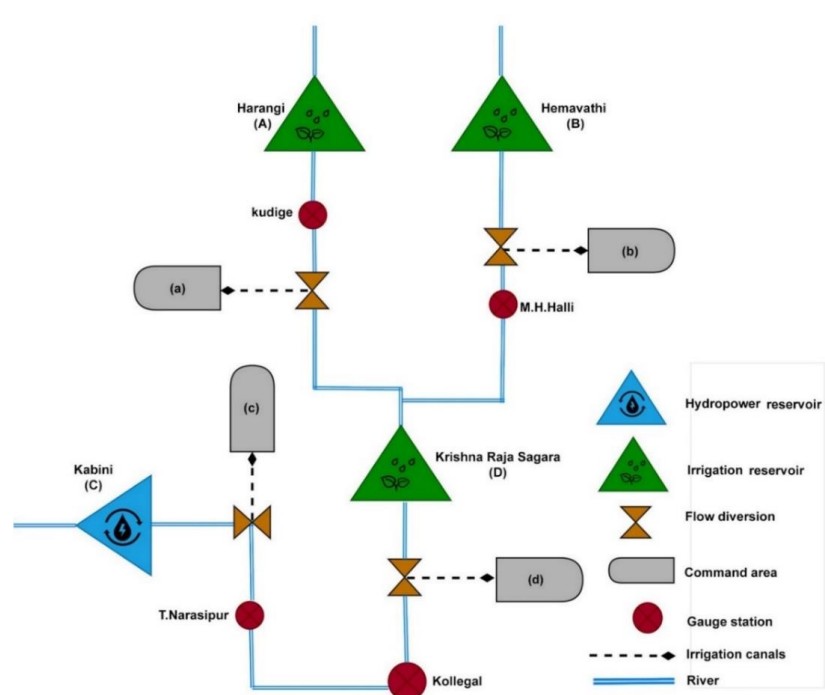


**Figure 5. Schematic representation of connections between reservoirs along the Upper Cauvery River**

**network**



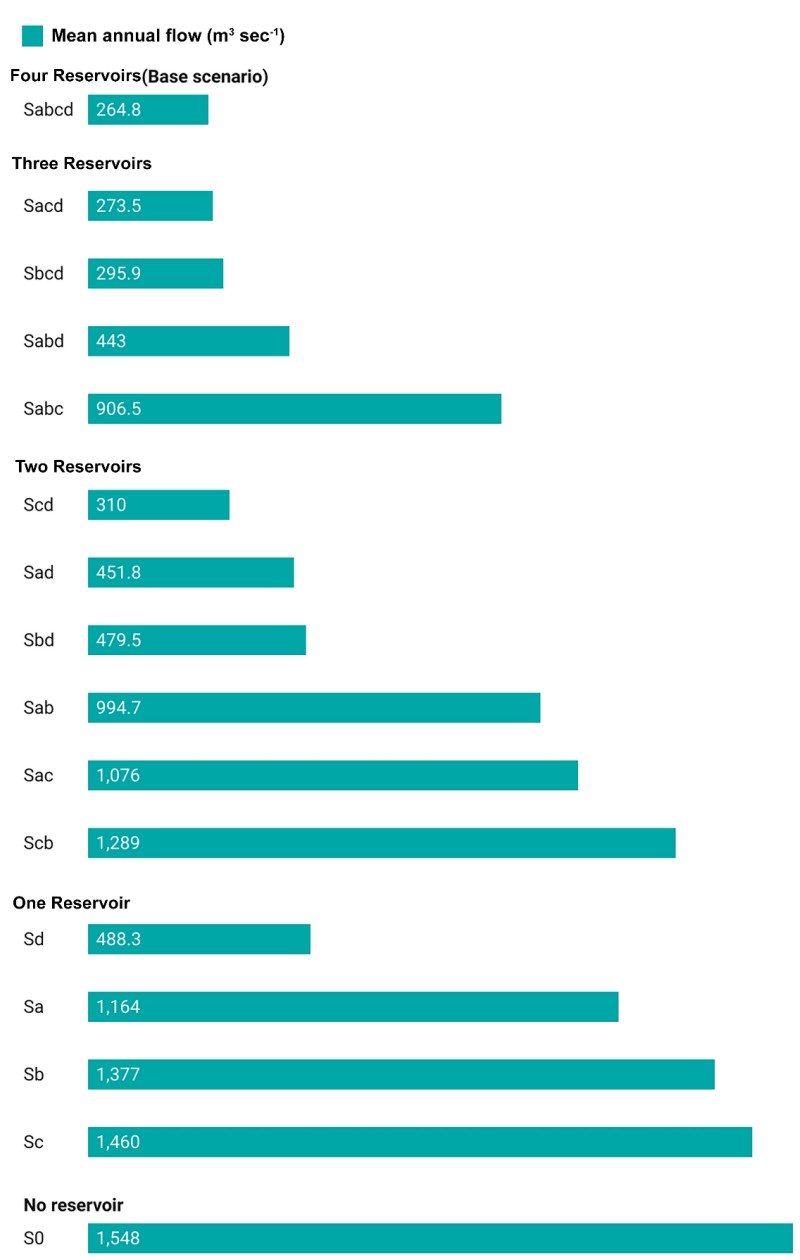

Figure 6. The mean annual flows resulting from different combinations of reservoirs



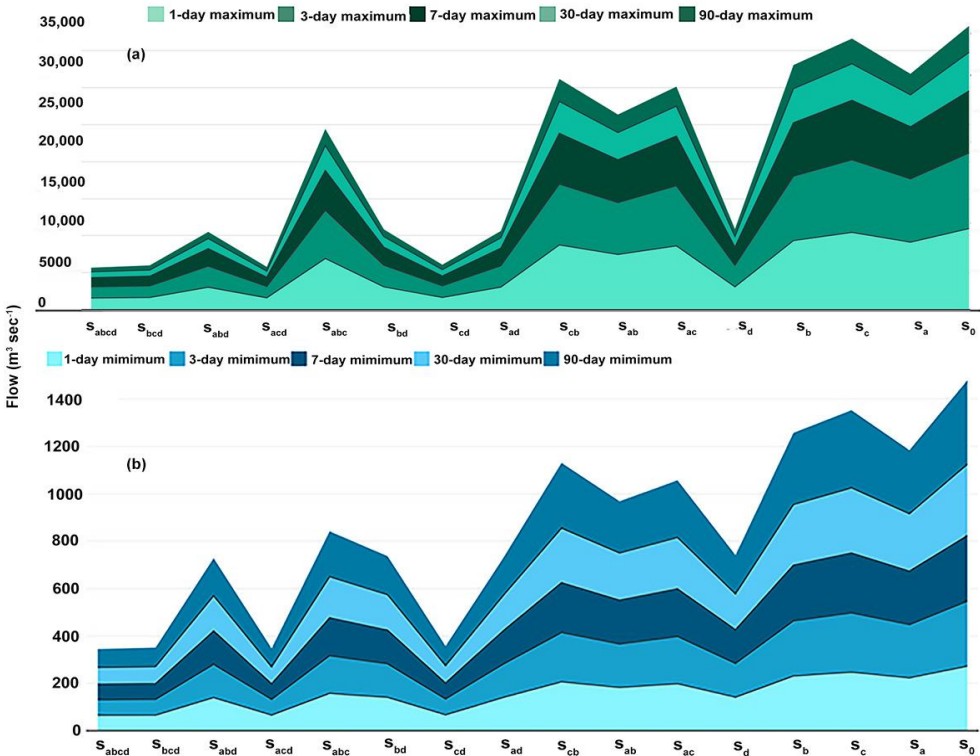

**Figure 7. The magnitude of annual extreme flow conditions of flow regimes generated by different combinations of reservoirs**






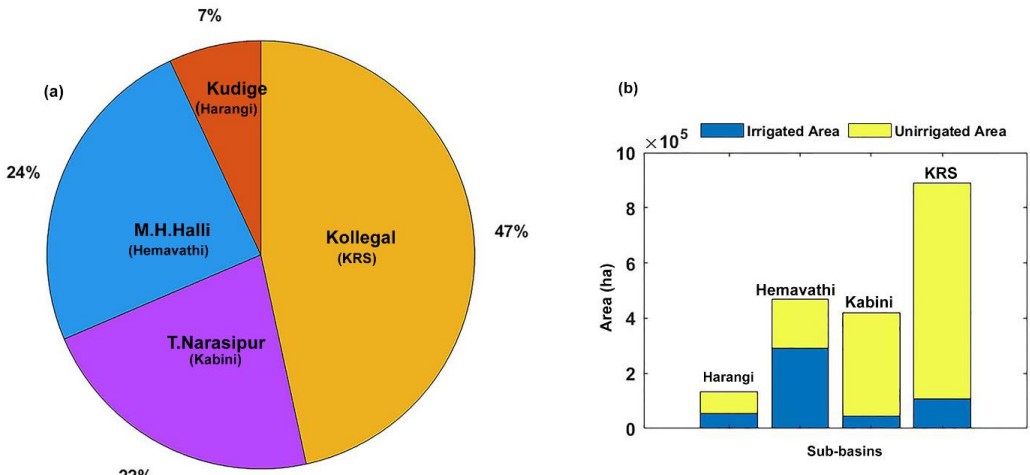

Figure 8. Overview of cultivated areas in different sub-basins. (a) represents the contribution of sub-basins to the total cultivated area of the Upper Cauvery basin, and (b) represents irrigated and unirrigated (or rainfed) areas in each sub-basin

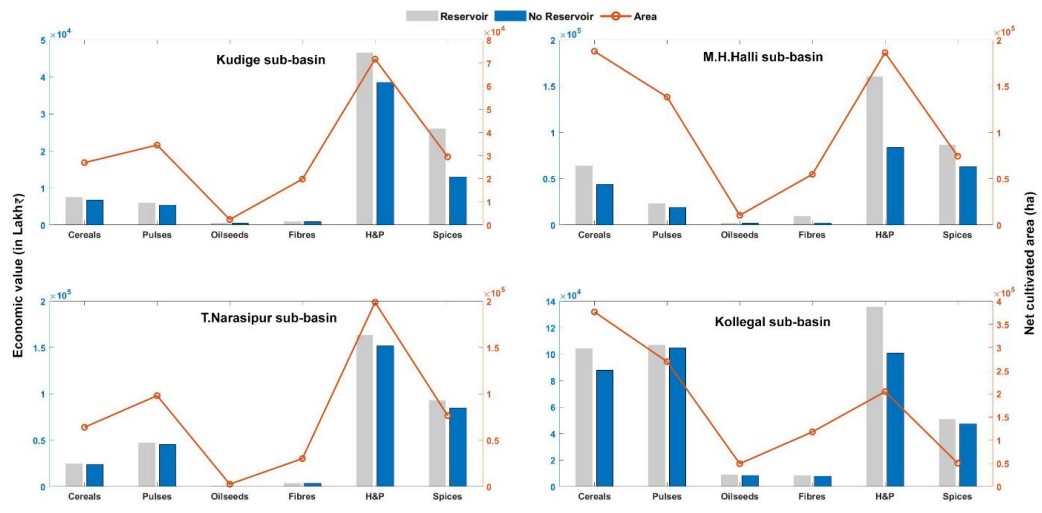

Figure 9. The economic value (Lakh ₹ per year) of different crop groups of individual sub-basins with and without reservoirs

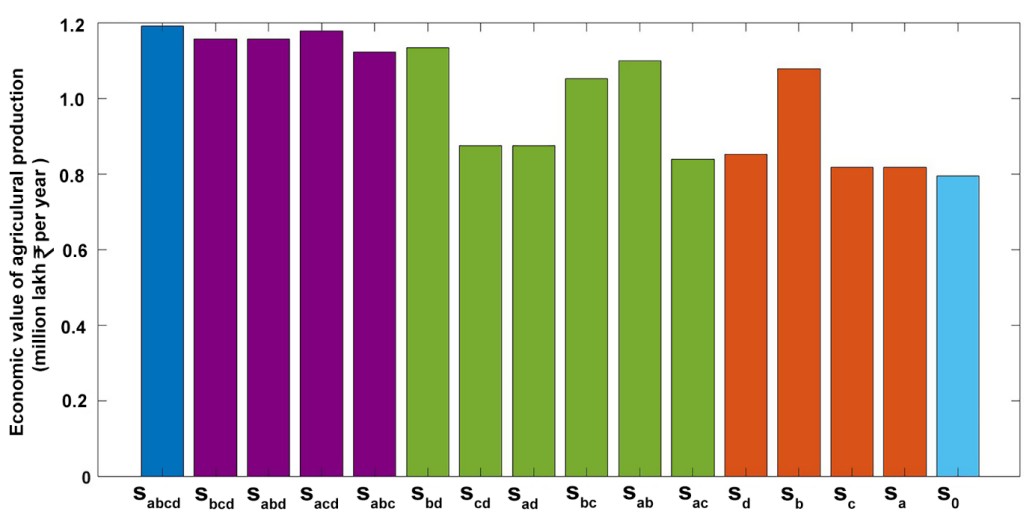


**Figure 10. The economic value of agricultural production under different scenarios of reservoirs**

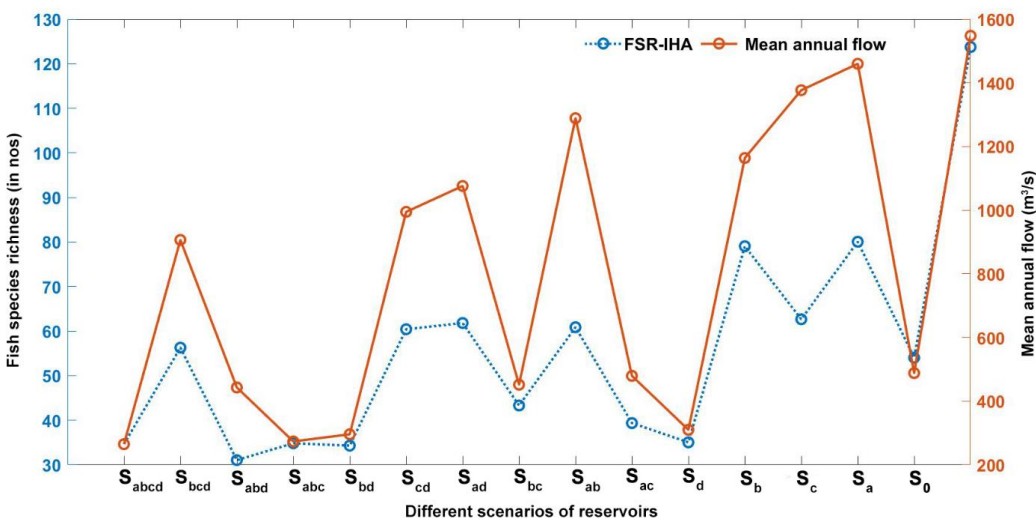

**Figure 11. The fish species richness (FSR-IHA) of the different combinations of reservoirs was calculated**

**based on mean discharge and flow regime characteristics**





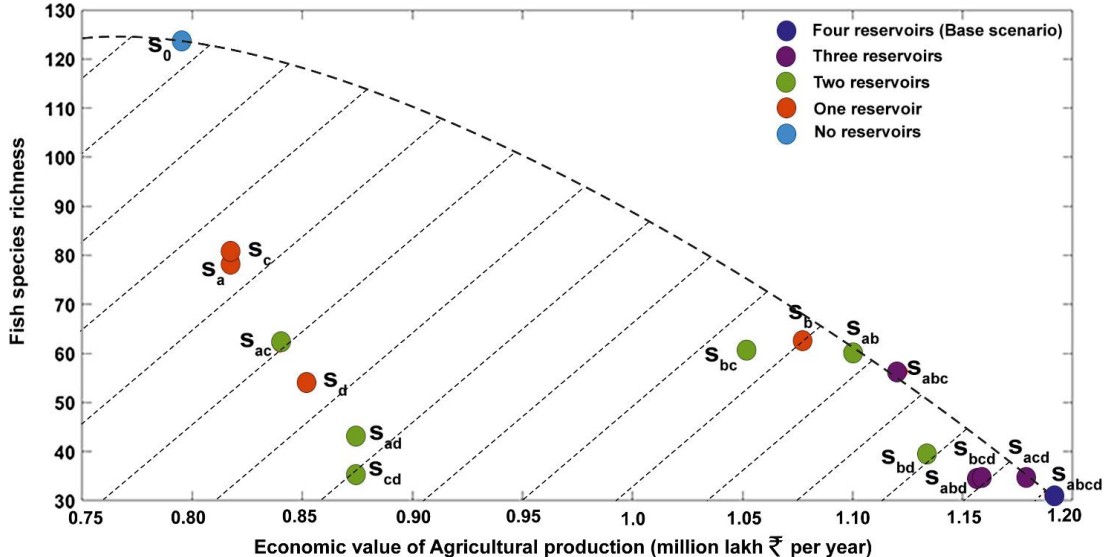

**Figure 12. Illustration of production set and production possibility frontier (PPF). The PPF is the outer edge**

**of the set, between agricultural production and fish species richness. The production set is generated as the**

**convex hull of FSR - value of production pairs**