# Peer review of "How economically and environmentally viable are multiple dams? A hydro-economic analysis using a landscape-based hydrological model at the basin scale"

_Hydrology and Earth System Sciences, 2023_

## Author Comment (AC1)

Dear reviewers, thank you for your valuable feedback. We have taken into account the insightful suggestions provided, which have allowed us to elaborate further on the content and improve the overall quality of the paper. The responses to the comments are presented below in a two-column format.

| General Comments | Response |
|---|---|
| The model should be properly described. | The aim of the paper is to assess the hydrological, ecological, and economic consequences of multiple dams within the study basin. To achieve this objective, a landscape based hydrological model (FLEX-Topo) was first developed and integrated with a reservoir model. The setup of this model was explained in detail, including its inputs, parameters calibrated and calibration results, in Ekka et al. (2022) and are summarized below, for the benefit of the editor and reviewer; but as these have already been published, these cannot be reproduced in the present paper, as this would constitute self-plagiarism. A brief description is provided below and it has been included in the revised manuscript.

"This integration involves modeling the operations of the reservoirs, as well as the hydrology of the upstream and downstream areas. By combining these models, the impact of reservoirs on the observed downstream flow patterns and the delivery of ecosystem services can be evaluated. The study focuses on four sub-basins, which are defined by the four gauging stations. Each sub-basin is further divided into two sections, representing the upstream and downstream areas of the respective reservoirs. The FLEX-Topo models for the upstream and downstream areas of a reservoir, respectively, contribute to the flow at a gauge station. Each sub-basin consists of reservoir model and command area associated with each reservoir. The flow at the downstream gauge station is measured and calibrated. Foe each sub-basin with reservoir, the outflow from upstream area becomes the inflow for the reservoir, and the outflow from the reservoir model enters downstream area. The outflow from downstream area is then calibrated at the gauge station. If the reservoir is removed, the outflow from upstream area is combined with the outflow from downstream area, forming the outflow at the gauge station. Afterwards, all the reservoirs are integrated to assess the effect of reservoirs on the flow downstream for varying configurations of all the reservoirs." |

| What are the input and output of the model? | The model inputs consist of two parts: Hydrological Response Units (HRUs) and forcing data. A detailed explanation of these components has been included in the revised manuscript as follows:

**"Creation of the Hydrological Response Unit (HRU)**
The creation of hydrological response unit (HRU) involves utilizing Digital Elevation Model (DEM), slope, and Height Above the Nearest Drainage (HAND) to categorize the landscape into distinct classes. In the ArcGIS software, the slope and HAND data are processed using an 80-meter resolution DEM. The delineation of the watershed area of the sun-basin is determined based on the gauge location, while for upstream, the watershed area is defined by the location of the dam. Subsequently, downstream area is obtained by extracting the upstream area from the entire watershed. The datasets containing DEM, slope, HAND, and basin boundaries are then clipped and exported to Matlab for further analysis. To differentiate between the three landscape classes, specific thresholds are selected. Areas with HAND greater than 5 meters and slope less than 11 percent are classified as plateaus. Areas with HAND greater than 5 meters and slope greater than 11 percent are identified as hillslopes. Finally, areas with HAND less than 5 meters are designated as wetlands, following the approach proposed by Gharari et al. (2011). These classified maps are then compared with land use maps to determine five distinct Hydrological Response Units (HRUs): Hillslope forests, Hillslope crops, Plateau forests, Plateau crops, and Wetlands. The determination of these HRUs is based on the proportion of landscape classes in the upstream area and downstream areas of the reservoir for each sub-basin. These HRUs are subsequently used to run the Flex-topo model.
**Forcing data:**
Rainfall and potential evapotranspiration data are utilized as inputs for the analysis. Daily gridded rainfall data with a resolution of 0.25° x 0.25° and temperature data with a resolution of 1° x 1° are obtained from the Indian Meteorological Department, Government of India (Pai et al., 2014; Shrivastava et al., 2009). These data are extracted for each sub-basin and used to drive both the FLEX-Topo model and the reservoir model. The potential evapotranspiration (ETo) is calculated based on the Hargreaves equation (Hargreaves and Samani, 1982), which considers the maximum, mean, and minimum temperature values. Runoff data for the gauges shown in Figure 2b is |

| | obtained from the Central Water Commission, Government of India. Information on the reservoirs including inflow, outflow, and storage levels, is accessed from the Karnataka State Natural Disaster Monitoring Centre, Government of Karnataka, India, through their official website (https://www.ksndmc.org/Reservoir_Details.aspx). Modelled discharge and actual evapotranspiration is obtained as output from the model which are used for further analysis." |
|---|---|
| How is the model simulated? | The hydrological model simulation is described and included in the revised manuscript as follows: "The present study utilizes a hydrological model called FLEX-Topo, which is based on topography and is used to simulate the behavior of river landscapes. The interaction between vegetation, soil, and climate is influenced by topography and plays a crucial role in shaping the characteristics of these landscapes. The FLEX-Topo model employs a catchment-scale simulation approach by dividing the landscape into distinct hydrological response units (HRUs), such as wetlands, hillslopes, and plateaus (Gharari et al., 2014). This parsimonious modeling approach has demonstrated its ability to be applied in data-scarce basins, as its structure is constrained by topography, requiring fewer calibration parameters, and yielding reliable flow simulations even under changing land-cover conditions (Gao et al., 2014; Savenije, 2010). Five HRUs are then determined based on the percentage of landscape classes for the upstream and downstream areas of the reservoir for each sub-basin. Afterwards, the model is calibrated using rainfall and potential evapotranspiration as forcing data". |
| It lacks basic explanation of what the variables and parameters are.  Which parameters are calibrated? | There are in total 25 parameters as indicated in Table 1 taken from Ekka et al. (2022). These are calibrated using Elitist Non-Dominated Sorting Genetic (NSGA-II) algorithm. This table has been included in the supplementary materials. Table 1. Model parameters prior ranges. These define the feasible range within which parameters are calibrated. |

| Parameters | Parameter Range | | | | |
|---|---|---|---|---|---|
| | Plateaucrop | Plateau forest | Hillslope crop | Hillslope forest | Wetlands |
| $I_{max}$ [mm/day] | 1-8 | 6-10 | 1-8 | 6-10 | 1-5 |

| Parameter | | | | | |
|---|---|---|---|---|---|
| (Storage capacity of the Interception reservoir) | | | | | |
| $C_e$ [-] (Fraction of $S_{u, max}$) | 0.1-1 | 0.1-1 | 0.1-1 | 0.1-1 | 0.1-1 |
| $Su_{max}$ [mm] (Maximum soil moisture capacity in the root zone) | 100-500 | 100-1000 | 100-500 | 100-1000 | 10-100 |
| $\beta$ [-] (Spatial heterogeneity in the catchment/shape parameter) | 0.1-5 | 0.1-5 | 0.1-5 | 0.1-5 | 0.1-5 |
| Pmax [-] (Maximum percolation rate) | 0.1-5 | 0.1-5 | - | - | - |
| D [-] (The splitter) | - | - | 0-0.5 | 0-0.5 | - |
| $C_R max$ [mm/day] (Capillary rise) | - | - | - | - | 0.01-1 |
| $K_f$ [d] (Recession coefficient of the fast reservoir) | 0.005 -1 | 0.005 -1 | 0.005-1 | 0.005-1 | 0.005-1 |

| **Catchment parameters** | |
|---|---|
| $K_s$ [d] (Recession coefficient of the slow reservoir) | 0.0001-0.01 |
| $T_{lag}$ [d] (Time lag between the storm and peak flow) | 0.1 – 30 |
| Frac 1 [-] (Fraction of forests cover) | The value is fixed (0 -1) based on the percentage of forest area in the sub-basin |
| Frac 2 [-] (Fraction of Irrigation) | The value is fixed (0 -1) based on the percentage of Irrigated area in the sub-basin |

Source Ekka et al., 2022

| The calibration results are not presented. | The results of the calibration and validation after integration of all the reservoirs and measured at the last gauge station is presented in Table 2 taken from Ekka et al., (2022). |
|---|---|
| | Table 2. The calibration and validation after integration of all the reservoirs |

| Model performance | -NSE [range] | MAE [range] $(10^6 \text{ m}^3 \text{ day}^{-1})$ |
|---|---|---|
| Reservoir Calibration (2011-2016) | -0.68 [-0.67 - (-0.69)] | 0.71 [0.70 - 0.72] |
| Flex-Topo Calibration (1991-2010) | -0.53 [-0.54 - (-0.52)] | 0.92 [0.92 -0.97] |
| Flex-Topo Validation (2011-2016) | -0.50 | 0.86 |

Within parentheses, the Pareto front ranges produced by the NSGA II algorithm are given for both -NSE and MAE. The MAE is always non-negative, and a lower value means a better prediction. The MAE value was recorded in the range of 0.70 - 0.72 $(10^6 \text{ m}^3 \text{ day}^{-1})$ which is in the acceptable range. Similarly, the -NSE value was observed between -0.51 to -0.73. The -NSE value less than -0.50 is acceptable (Table 2). Kindly note our response to the misunderstanding about our reporting of NSE. We had reported negative NSE values instead of positive NSE (due to its use to maximize as an objective in NSGA II). NSE values are greater than 0.5 and sometimes even around 0.7, which indicates reasonably good performance of a model at daily scale that also incorporated reservoir operations. In the current model, the negative reverse of NSE value is used to calibrate and validate the model parameters.

| How the flow regime was generated, and which time frame was considered for simulation is unclear. | The missing information on model simulation is indicated in the revised manuscript as follows:

"To calibrate and validate the FLEX-Topo models, the dataset of topographic maps, rainfall, and potential evapotranspiration was used. Specifically, the dataset from January 1991 to December 2010 was used for calibration, |
|---|---|

| | and the dataset from 2010 to 2016 was used for validation. The reservoir models were calibrated using the dataset composed of inflow, outflow, storage, rainfall, and potential evapotranspiration, for the reservoirs covering the period from January 2011 to December 2016." |
|---|---|
| The overall approach is to compare scenarios of current state of 4 reservoirs vs. not having one or more reservoirs. From rather planning perspective, isn't the more important question is if the smaller multiple reservoirs are better or worse than single or fewer larger reservoirs given all the water demand needs are met? Please mention what are your research questions. | The reviewer makes a very valuable suggestion, namely to consider also the alternative of replacing large reservoirs by one or more smaller reservoirs. While this is a valuable question for future research, it falls outside the scope and objective of the present paper. The objective has been refined in the revised manuscript to provide a more precise description, resulting in a better understanding of the intended goal as follows:

"This paper aims to answer how different combinations of spatially located reservoirs of varying sizes in the Upper Cauvery River basin perform both in terms of economic value, as measured in terms of agriculture production, and environmental performance, as measured in terms of fish species richness. In this sense the paper evaluates the most suited configuration of existing reservoirs that effectively meets water demand needs while considering environmental sustainability." |
| The agriculture production and environmental consequences are currently tied to regulated water and scenarios of the number of reservoirs, but it seems like reservoir operations and seasonal influence of streamflow is overlooked in result analysis. As it mentions "embedded reservoir model", I believe it is incorporated in the model, but it is not clearly described in manuscript. | The following paragraph is added to the revised manuscript to explain the reservoir operations and seasonal influence of streamflow

"A hydrological model based on the landscape is developed and integrated with a reservoir model. This integration involves modeling the operations of the reservoirs, as well as the hydrology of the upstream and downstream areas. The model runs at daily time steps, so reservoir operations and streamflow are simulated at daily time steps. Thus, seasonality is implicitly considered." |
| Please define fish species richness. | The fish species richness is defined and this has been added in the revised manuscript as follows: |

| | "Fish species richness refers to the number of different fish species present in a particular area or ecosystem. It is one of the measures of biodiversity and represents the diversity and abundance of fish species within a given habitat or geographical region. Species richness is commonly used to assess the ecological health and complexity of aquatic ecosystems. The higher the fish species richness, the greater the variety of fish species present in an area, indicating a more diverse and potentially healthier ecosystem. Fish species richness can be influenced by various factors, including habitat type, water quality, temperature, and availability of food and resources. Monitoring and understanding fish species richness are important for conservation efforts, ecosystem management, and assessing the impacts of human activities on aquatic environment."

 The analysis does not focus on a specific fish species but rather considers the presence of various fish species within a riverine ecosystem. This approach is further clarified and added to revised manuscript as follows: |
|---|---|
| What types of fish species are considered here? | "The analysis focuses on fish species richness, which refers to the number of different fish species present in a specific area or ecosystem. As a result, no specific fish species is targeted in this study. By considering the collective presence of various fish species, the analysis provides insights into the overall ecological health and biodiversity of the system under investigation. However, the ichthyofauna diversity of the river Cauvery has recorded 146 fish species belonging to 52 families (Koushlesh et al., 2021). From an environmental conservation point of view, the Cauvery is critical to the habitat for many fish species. For example, the Mahseer population which is known as hump-backed mahseer (Tor remadevii) is listed as endangered on the IUCN red list and is found only in the Cauvery basin (Pinder et al., 2015; Sreenivasan et al., 2021). Along the river, many old fish sanctuaries are considered sacred which has the potential to contribute significantly to ecological health and biodiversity conservation." |
| Just the number is considered or their migration pattern as well. | The fish migration pattern has not been studied due to many limitations which includes tracking efficiency, sample bias, limited spatial coverage, as well as species-specific challenges. Tracking fish migration often relies on the use of electronic tags, such as acoustic or satellite tags, which may have limitations in terms of accuracy and precision |

| | (Planque et al., 2011, Elsdon et al., 2008). Fish migration studies typically focus on specific areas or migratory routes due to logistical constraints. This limited spatial coverage may overlook other migration pathways or key habitats. Different fish species exhibit diverse migration behaviors and ecological requirements, which present unique challenges for studying their migration patterns. It would be difficult to incorporate all these parameters in the current hydrological model, and is beyond the scope of the present article. |
|---|---|
| Is the fish species richness as an indicator enough to describe ecosystem health. | Fish species richness as an important indicator to describe ecosystem health. It has been explained and included in the revised manuscript as is follows:

"Aquatic ecosystem health serves as a comprehensive reflection of the physical, chemical, and biological integrity of river ecosystems, the overall structure and function of the aquatic ecosystem (Chen et al., 2019; Aazami et al., 2019). Previous studies have investigated various factors to identify the key determinants influencing river ecological health, including benthic macroinvertebrates, river habitat conditions, and water quality parameters (Chen et al., 2019). However, when considering the biological indicators, fish health becomes crucial as it directly links to provisioning services like food and human health. Therefore, fish species richness is chosen as an important indicator of river health, reflecting the overall health and diversity of the aquatic ecosystem." |
| Why was the fish species richness only based on the empirical function, and how the equation 3 was adopted and validated for this study basin?

Given the size of the basin, does a single FSR value for the whole river networks adequate? | The estimation of fish species richness was based on a global statistical model developed by Iwasaki et al. (2012). However, the same formula is validated in 84 major basins worldwide by Yoshikawa et al. (2014). The following paragraph is added to the revised version of the manuscript.

"The FSR (Fish Species Richness) value is derived by calculating the species richness of major basins worldwide (Yoshikawa et al., 2014; Iwasaki et al. (2012). The value obtained from the equation presented by Iwasaki et al. (2012) is centred in the 20-250 range. Other field studies have confirmed that the FSR in cauvery river basin tends to be around 146 species (Koushlesh et al., 2021). In the current study, the estimated FSR for the Cauvery River basin ranges from 70 to 123 species under different projected scenarios. This range of values provides sufficient validation for the results obtained. |

| | Kindly note that the primary objective of using FSR is not to predict FSR values for the basin, but rather to demonstrate how the characteristics of the river basin and its flow can impact fish species richness and different choices of the configuration of the reservoirs can lead to be different economic values and (fish) biodiversity in the long run (since we are using averages of these two variables over 16 years). This, in turn, affects the overall biodiversity of the river ecosystem and subsequently leads to a decline in river sustainability and resilience. By assessing and understanding these relationships, it becomes possible to identify the potential impacts of flow alterations and basin modifications on the long-run biodiversity and ecological stability of the river systems". |
|---|---|
| Are there any other indexes which was not used in this study due to any limitations? Any such limitations should be mentioned. | Direct field sampling from the river streams is the best method for developing indices for this study however this approach was not pursued due to cost implications associated with sampling. The following paragraph is added to the revised manuscript to highlight the limitations of the study.

"Although directly sampling fish species from the streams and analyzing their richness in relation to river flow characteristics could have provided more conclusive results, this approach was not pursued due to cost implications associated with sampling of all major streams in the Cauvery basin. Hence, this method was intentionally omitted". |
| **Specific comments** | **Response** |
| Section 2.3: Please mention the quality of data obtained. | The paragraph is revised by providing the source of data obtained as follows:
"Rainfall and potential evapotranspiration data are utilized as inputs for the analysis. These are daily gridded rainfall data with a resolution of 0.25° x 0.25° and temperature data with a resolution of 1° x 1° are obtained from the Indian Meteorological Department, Government of India (Pai et al., 2014; Shrivastava et al., 2009), which are generally considered of good quality. Runoff data is obtained from the Central Water Commission, Government of India. Information on reservoirs, including inflow, outflow, and storage levels, is accessed from the Karnataka State Natural Disaster Monitoring Centre, Government of Karnataka, India, through their official website (https://www.ksndmc.org/Reservoir_Details.aspx). Both data sets are considered to be of good quality" |

| | |
|---|---|
| If there are any data gaps, if so, how the data gaps were addressed? | The data gaps are described and included in the revised manuscript as follows: "For reservoir model calibration only a time series of six years of daily inflows, storage and outflows was accessible. However, extended periods of streamflow data for the corresponding gauges, rainfall and temperature data for the basins were available. Thus, the six-year reservoir data was used to calibrated the reservoir models and the other streamflow and input forcing data were utilized to calibrate the integrated FLEX-Topo and reservoir models. Finally, since a standardized operating rule curve was applied to all reservoirs (its parameters calibrated for each reservoir), it is acknowledged that the specific water discharges of certain dams might not have been accurately captured by the reservoir model. The validation results also indicate a bias in the modeled flows of two reservoirs. Therefore, enhancing the model calibration process may involve incorporating operating rule curves that consider specific reservoir functions and flow requirements.  which is beyond the scope of this paper. However, it should be noted that addressing this aspect is beyond the scope of the current paper." |
| Section 2.4: Please provide the parameters that are calibrated.

Also, please mention the population size, number of generations used, and crossover and mutation probability used and how those parameter values were determined? | The Elitist Non-Dominated Sorting Genetic (NSGA-II) algorithm is used to calibrate the model parameters (Deb et al., 2000).  NSGA-II is a multi-objective optimization algorithm. It simultaneously optimizes multiple objectives by identifying parameters that yield model performances that are not dominated by any other feasible parameters in the multi-objective space (Efstratiadis & Koutsoyiannis, 2010).

In total 25 parameters were calibrated and are indicated in Table 1 of Ekka et al. (2022; reproduced above). The population size, number of generations, crossover and mutation probability were indicated in Table 3.

The following paragraph is added to the revised manuscript to provide a detailed explanation:

"Two objective functions are defined and minimized simultaneously. The first objective ($f_1$) is the negative of Nash-Sutcliffe Efficiency (NSE) and the second objective ($f_2$) is the Mean Absolute Error (MAE). Please note by minimizing the negative of NSE, we are in effect maximizing NSE while minimizing MAE. |

$$f_1 = -NSE = -1 + \frac{\sum_{i=1}^{n}(Q_i^m - Q_i^o)^2}{\sum_{i=1}^{n}(Q_i^o - \bar{Q}_o)^2}$$

$$f_2 = \text{MAE} = \frac{1}{n}\sum_{i=1}^{n}|Q_i^o - Q_i^m|$$

Here, $Q_i^m$ is the $i^{th}$ observation for the observed discharge being evaluated. $Q_i^o$ is the $i^{th}$ value of the modelled discharge. $\bar{Q}_o$ is the mean of observed discharge and $n$ being the total number of observations. Units of discharge are mm/day (which are volumetric flow rates divided by the upstream flow contributing areas). In case of calibrating the reservoirs, $Q_i^m$ is the observed outflow from the reservoir and in the case of calibrating the integrated model, it is the flow measured the corresponding streamflow gauge downstream of the reservoir.

The parameter sets calibrated for the FLEX-Topo model and the reservoir model are provided in the supplementary materials. The NSGA-II parameter setting may have different impacts on computational effectiveness. The population size, number of generations were indicated in Table 3. The population crossing over and population mutation play critical roles during optimization higher fraction of the population crossing over (0.9) and a lower value of mutation value are preferred for better convergence and to prevent the population from getting trapped in local optima (Wang et al., 2019). The population size depends on the number of the decision variables calibrated in the model and keeping the population size five times the number of decision variables is considered ideal for the simulation (Gutierrez et al., 2019). Since for Flex-Topo, there are 20 parameters, the population size is kept at 100. Similarly, for the reservoir model, the number of parameters is five, which translates into a population size of 25. Higher population sizes were also attempted but not used and reported for later analysis because the performance achieved was similar to the reported population sizes. The number of iterations is first tested using 50, 100, 250 and 500 iteration runs and 250 was finally chosen based on the best optimization results".

| Table 3. Parameter setting for NSGA II optimization of the model | | |
|---|---|---|
| **NSGA parameters** | **Reservoir calibration** | **Integrated FLEX-Topo Calibration** |
| No. of Iterations | 250 | 300 |
| No. of decision variables | 5-8 | 25 |
| No. of population size | 25-40 | 125 |
| Population Crossover | 0.7 | 0.7 |
| Mutation probabilities | 0.2 | 0.2 |
| New generation selection | Elitist selection | Elitist selection |
| Ordering criteria | Crowding distance | Crowding distance |
| Source: Ekka et al., 2022 | | |

| | |
|---|---|
| Line 162 mentions it is calibrated based on downstream streamflow, but line 172 mentions target range of metrics with unit as mm/day. Any reason for not using the flow unit here? Why only the downstream flow was compared, is it because inflow was a model input? It is not very clear. | An explanation is provided and it is included in the revised manuscript as follows:

 The Flex-Topo model operates on a daily time scale, and it relies on forcing data expressed in mm/day. Consequently, when calibrating the model's simulated discharge, the observed discharge in cubic meters per second (cumec) is converted into mm/day |
| Line 181-182: What is landscape model? Also, the sentence is confusing to read. | The landscape model is explained and included in the revised manuscript as follows:
 "Topography plays a crucial role in shaping the river landscape, as it evolves in conjunction with vegetation, soil, and climate (Savenije, 2010; Gao et al., 2014). Consequently, it governs the dominant hydrological processes within a catchment (Gao et al., 2014) and serves as a valuable constraint for determining and transferring model structures across space and time (Gao et al., 2016; Nijzink et al., 2016). The landscape model (eg. Flex-Topo) effectively |

simulates catchment responses by dividing them into distinct hydrological response units (HRUs), such as wetlands, hillslopes, and plateaus (Gharari et al., 2014) based on the catchment structure and processes."

| | |
|---|---|
| Figure 5: It looks like station T. Narasipur is along the same channel upstream of Kollegal as per Figure 3, although reservoir C is on tributary. If it is on tributary, it should be mentioned somewhere to avoid confusion. | The figure 3 has been modified to indicate the exact location of gauge stations on the map. Reservoir C is located on tributary and the gauge station T.  Narasipur is also located on tributary below Kabini reservoir (C).

[Figure]

Figure 3, An overview of the study area. The reservoirs in the study area are labelled as A, B, C, and D, representing Harangi, Hemavathi, Kabini, and KRS reservoirs, respectively. The labels CA, CB, CC, and Cd are used to denote the respective command areas associated with these reservoirs |

| | |
|---|---|
| What is the significance of the hydrological sub-basin watershed delineation in this study with regards to hydrology since it is not a physical based model. | The significance of the hydrological sub-basin watershed delineation in this study is explained and included in the revised manuscript as follows:

"The delineation of sub-basins is fundamental for conducting hydrological modeling. In this study, each reservoir is associated with a gauging station, which serves as a watershed or sub-basin. Moreover, each sub-basin is divided into upstream and downstream sections, primarily determined by the location of the dam. The classification of landscape classes for creating hydrological response units is based on the specific areas of the upstream and downstream regions. And therefore, the delineation of watersheds plays a critical role in the present study, as it provides the necessary framework for accurately defining hydrological response units based on the spatial characteristics of the upstream and downstream sections." |
| How the watersheds are delineated – is it using ArcGIS Hydrology tools, please mention that? | The delineation of the watershed is explained and included in the revised manuscript as follows:

"The delineation of the overall watershed area is based on the gauge locations to calibrate the integrated hydrological model with the embedded reservoir model (refer figure 2a and figure 3). The watershed area for F1 is determined by the dam location. To obtain F2, the F1 area is extracted from the entire watershed. The classification of topographic features such as Digital Elevation Model (DEM), slope, and Height Above the Nearest Drainage (HAND) is conducted based on the size of F1 and F2 to categorize the landscape into three main classes. In ArcGIS, slope and HAND are processed using an 80-meter resolution DEM. The raster datasets containing DEM, slope, HAND, and basin boundaries are then clipped and exported to Matlab for further analysis." |
| Does the line 184 says the sub-basin for reservoir KRS is delineated with Kollegal station as outlet? But it looks like the reservoir is far upstream from this point | Yes, the sub-basin corresponding to KRS is delineated by the gauging station Kollegal as outlet. The line 184 is elaborated and included in the revised manuscript as follows:

"The sub-basin corresponding to KRS is delineated by the gauging station Kollegal (and hence the flows are modelled at this station), except those flows generated by contributing areas corresponding to gauge stations Kudige, M.H. Halli and T. Narasipur. Such calibrated models of flows (with or without respective reservoirs) at the gauge |

| | |
|---|---|
| | stations downstream of each of the 4 reservoirs, instead of observed flows, are then used for simulating flow regimes at the gauging station Kollegal for various possible configurations of reservoirs upstream." |
| Section 2.6: Which indicators are considered, and which are important for this study? | The importance of IHA has been indicated in the following paragraph and this has been included in the revised manuscript as follows:

"The Indicators of Hydrological Alteration (IHA), initially proposed by Richter et al. (1996), are used to measure the effects of different reservoir combinations on the flow regime in the Upper Cauvery basin. The IHA indicators are grouped into five categories based on monthly flow magnitude, annual extreme flow conditions, and the frequency and duration of high and low flow rates. Major indicators used in the study include mean annual discharge, low flows, high flows, low pulse rate, high pulse rate. These parameters have significant relationships with river ecosystems, which makes them suitable for assessing the impact of dams, barrages, and other water diversion structures on the flow regime." |
| Why the IHA was used? It would be useful to mention here. | The paragraph below discusses the practicality or benefits of IHA and this has been included in the revised draft as follows:

"Earlier methods of assessing the impact of impoundments on river channels have involved field surveys, statistical analyses (Yan, 2010), and geomorphic change detection tools (Wheaton, 2015). However, the Range of Variability Approach and the associated IHA framework provide a more systematic assessment of flow changes. The IHA method utilizes daily streamflow values and characterizes a flow regime based on factors such as magnitude, duration, frequency, timing, and rate of change of flows. Although the application of the IHA method has been relatively limited in studies of Indian rivers (Mittal et al., 2014, Kumar and Jayakumar, 2020, Borgohain et al., 2019), this study aims to use the IHA method to gain valuable insights into the impacts of major dams on the flow regime of the Upper Cauvery basin. By doing so, it contributes to a better understanding of the ecological consequences of water diversion and reservoir operations in the region" |

| Line 211 – 215: The introduction of PPF can be improved. Currently it seems inadequate, linking it to the objective of this study and why this tool/graph is ideal would be better. How was it generated should be explained in methods? I also think Figure 12 is a great visualization. | Thank you for the feedback. The introduction to production possibility frontier (PPF) has been revised and added to the revised manuscript linking it to the objective of the study as follows:

"The production possibility frontier (PPF), also known as the production possibility curve or boundary, is a graphical representation of the different combinations of goods or services that an economy can produce given its limited resources and technology (Martinez-Harms et al., 2015). It shows the maximum output of one good that can be produced in relation to the production of another good, given the existing resources and technology.

Since dams have the potential to impact ecosystems through habitat alteration, water flow regulation, and the disruption of natural processes, the PPF helps to assess the trade-offs and opportunity costs associated with the allocation of resources between the production of goods and the provision of ecosystem services. The PPF can illustrate the trade-off between the production of goods, often associated with economic development, and the provision of ecosystem services. It highlights that allocating resources towards dam construction may result in a reduction of certain ecosystem services (Blachly et al., 2023). For instance, the construction of a dam may alter the natural flow of a river, affecting downstream water quality and biodiversity. By analyzing the PPF, decision-makers can assess the optimal allocation of resources between dam construction and the preservation of ecosystem services. The PPF helps in identifying the point at which the trade-off between economic development and ecosystem services is balanced, ensuring sustainable resource use and minimizing negative environmental impacts (Gómez-Baggethun et al., 2010). Furthermore, the PPF can aid in evaluating the potential gains or losses in ecosystem services resulting from dam projects. It allows decision-makers to compare different scenarios and assess the opportunity costs associated with alternative resource allocations. This analysis assists in making informed decisions about dam construction, considering the ecological consequences and potential benefits of preserving or enhancing ecosystem services (Blachly, 2023)." |
| Section 3.1: The use of term "spatial configuration" is vague. Generally, it could | The term spatial configuration of reservoirs is appropriate for the study. An explanation is provided to justify the term and it is included in the revised manuscript as follows: |

| | |
|---|---|
| mean configuration in term of location, storage size, total number, or design configuration of reservoirs using an algorithmic approach. Currently it is not the case in this study, as it is just combinations of reservoirs, so it should be rephrased correctly. | "The "spatial configuration of reservoirs" refers to the arrangement or layout of reservoirs within a given geographic area. It involves the positioning, distribution, and connectivity of individual reservoirs in relation to each other and the surrounding landscape. The configuration can have significant implications for the hydrological, ecological, and social aspects of the reservoir system. The spatial configuration determines how water is stored, distributed, and managed within a hydrological system. Factors such as topography, river networks, land availability, environmental considerations, and engineering constraints influence the spatial configuration." |
| Section 3.1.1: What is the location in the catchment that the mean annual flow refers to in Figure 6? | The following explanation is provided and included in the revised manuscript as follows: "All the hydrological indicators are calculated based on the discharges that are simulated at the Kollegal gauge station corresponding to each reservoir spatial arrangement." |
| Line 282 – "The highest mean annual flow was estimated for S0 followed by Sc and Sb". This seems to be obvious since the flow in unregulated without any reservoir in S0 scenario and the contributing areas of the reservoirs in Sc and Sb scenarios makes up only a small portion of the catchment area. | The explanation is provided and included in the revised manuscript as follows: "It is acknowledged that S0, being an unregulated sub-basin without any reservoir, exhibits the highest flow due to the absence of flow regulation. In contrast, Sc, which is a hydropower reservoir, needs to release water regularly for electricity generation purposes. As a result, S0 is estimated to have the highest mean annual flow, followed by Sc and Sb." |
| Line 289 – "Sabd has less impact compared to Sacd despite Kabini (C) having less storage capacity compared to the Hemavathi reservoir (B)". | The following explanation is provided and included in the revised manuscript "Since the combination Sabd has Hemavathi reservoir which falls in the M.H. Halli sub-basin which receives highest rainfall thereby contributing significantly to the overall flow regime, Sabd has less impact compared to Sacd despite Kabini (C) having less storage capacity compared to the Hemavathi reservoir (B)." |

| | |
|---|---|
| The mean annual flow is not only the function of reservoir volume, but perhaps the contributing area, land use, water use, precipitation pattern (not sure what is the time span considered!) and other properties which was not addressed in the study and the results are poorly presented. | As correctly said, the mean annual flow is not only the function of reservoir volume, but also the contributing area, land-use, water use, precipitation pattern including temperature. All these are already factored in the landscape based hydrological model, which has been integrated with reservoir operation models, that is used for simulating the daily streamflow time series corresponding to each reservoir spatial arrangement.

In the methodology section, it is mentioned that a landscape-based hydrological model is employed. This model utilizes 25 parameters that are based on the hydrological response units (HRUs) formed through a classification process. The classification is performed using digital elevation data, Height Above the Nearest Drainage (HAND) information, slope data, and land use patterns. These parameters are essential in characterizing the hydrological behavior of different areas within the study area. By considering factors such as elevation, HAND, slope, and land use, the model can capture the variability in terrain, drainage patterns, and vegetation cover that influence the movement and distribution of water in the landscape

The model requires inputs of rainfall and potential evapotranspiration data to drive the simulation. Rainfall represents the input of water into the system, while potential evapotranspiration represents the loss of water from the system due to evaporation and transpiration processes. To ensure the accuracy and reliability of the model, it undergoes a calibration and validation process. This process involves comparing the simulated streamflow generated by the model with observed streamflow data from a gauging station located downstream. Note this is done before the model is used to simulate various possible reservoir spatial arrangement and its effect assessed at Kollegal gauge station. The performance of the model is evaluated using statistical metrics such as the negative Nash-Sutcliffe efficiency (-NSE) and negative mean absolute error (MAE). The -NSE measures the agreement between observed and simulated streamflow, with values closer to -1 indicating a better fit. The MAE provides an indication of the average difference between observed and simulated streamflow, with lower values indicating better model performance. |

| | |
|---|---|
| | This ensures that the model captures the essential hydrological processes and reproduces the streamflow patterns under various reservoir arrangements reasonably well. We therefore disagree with the reviewer's observation that this "was not addressed in the study". |
| Figure 6: It is redundant with Table 2 (Mean annual flow column). Presenting hydrographs would be more informative than Figure 6 since there are only 16 scenarios and expanding the discussion on the role of reservoirs and seasonal streamflow. | The Table 2 is deleted, and the figure is modified to hydrograph as suggested.

[Figure]

Figure 6.  The mean annual flows resulting from different combinations of reservoirs

The discussion on role of reservoirs and seasonal streamflow has been expanded and included in the revised manuscript as follows:
"Reservoirs play a crucial role in the management and control of seasonal streamflow. They offer a range of advantages by storing excess water during periods of high flow and subsequently releasing it during drier periods. |

| | However, the decrease in summer flows is observed in all combinations involving a large reservoir (D), as evidenced by the model results. This reduction in flow can have negative consequences on the downstream aquatic habitats, as well as the migratory and reproductive behaviors of fish species. The migratory behavior of fish during the spawning season relies heavily on the frequency and duration of high and low flow pulses. Wang et al. (2016) highlight the critical role of these flow patterns in supporting fish migration. Several studies have also indicated that the construction of reservoirs, leading to reduced flows and alterations in natural flow pulses, poses a threat to the survival of migratory fish species like *Tor pitutora* in the Cauvery basin (Pinder et al., 2015)." |
|---|---|
| Section 3.1.2: Line 300-310-Similar to the above comment, the presentation of the result is very obvious because KRS (D) has the largest capacity and any combination with $S_D$ result is lowest annual extreme flow with the exception for $S_C$ which has comparative higher outflows being the only hydroelectric reservoir. | An elaborate explanation is provided and it is included in the revised manuscript as follows:

"KRS (D) has the largest capacity and any combination with $S_D$ result is lowest annual extreme flow with the exception for $S_C$ which has comparative higher outflows being the only hydroelectric reservoir. The comparison of a scenario with two irrigation reservoirs and one hydropower reservoir (Sabc) to a scenario with two irrigation reservoirs (Sbd) indicates that the former has less impact on mean annual extreme flow conditions such as 1, 2 and 7-day minimum than the latter. Comparing similar combinations of two reservoirs only for irrigation (Sad and Sbd) versus those that contain the hydropower reservoir (Scd) indicates that the hydropower reservoir decreases the low pulse count and low pulse duration compared to irrigation reservoirs. It indicates that the inclusion of a hydropower reservoir in a scenario with irrigation reservoirs can have a mitigating effect on mean annual extreme flow conditions, helping to maintain higher minimum flow levels during critical time periods. However, it also highlights that the presence of a hydropower reservoir may impact the frequency and duration of low flow pulses more than scenarios without hydropower reservoirs. These findings emphasize the importance of considering the specific characteristics and objectives of different types of reservoirs when evaluating their impacts on flow dynamics and designing sustainable water management strategies." |

| | |
|---|---|
| Table 2: It is not clear what are the other hydrological and environmental indicators besides mean annual flow. Perhaps it is better to define it somewhere. | The hydrological and environmental indicators are defined as suggested and included in the supplementary materials as follows: |

The second column of the first row contains the following table:

| |
|---|
| Mean annual flow (m3/s): Median ($m^3s^{-1}$) of daily flow condition from January to December |
| Low pulse count (days): No of times in a year when the flow is lower than the 25 % percentile of the flow period in analysis |
| High pulse count (days): No. of times in a year when the flow is higher than the 75 % percentile of the flow period in analysis |
| Low pulse duration (days): The median duration of the low pulses (days) |
| High pulse duration (days): The median duration of the high pulses (days) |
| Extreme low peak: Minimum flow event during each water year or season |
| Extreme low frequency: Frequency of extreme low flows during each water year or season |

| | |
|---|---|
| Line 306 – 310: A reference to figure would be useful here. | Reference to figure is added in the line 306-310 in the revised manuscript as indicated below: "The comparison of a scenario with two irrigation reservoirs and one hydropower reservoir (Sabc) to a scenario with two irrigation reservoirs (Sbd) indicates that the former has less impact on mean annual extreme flow conditions such as 1, 2 and 7-day minimum than the latter (Figure 7)." |
| Section 3.1.3: Line 324-333 – Residence time is briefly introduced for the first time here without introducing reservoir operations at all. Also, till here the description was based only on the size of reservoirs. It seems there are significant seasonal effects of streamflow and reservoirs operation which is overlooked. Further it describes A is significantly small in | The operating rules of all the reservoirs have been calibrated and are embedded in the integrated model that is then used to simulate flow regimes of various reservoir spatial arrangement. Since the model runs at daily scale, the seasonality and operational effects on streamflows have been incorporated. These are then analysed through various hydrological and environmental indicators mentioned above.

"The residence time for all the reservoirs has been calculated separately as indicated in Table 4. Generally, reservoirs with longer residence times tend to have a larger impact on the flow regime compared to reservoirs with a smaller residence time. However, $S_A$(with Harangi reservoir) has higher impact on the flow regime than $S_B$ (with Hemavathi |

capacity than B, A has short residence time than B, and the unregulated flow in B is less than in A, however $S_A$ has higher impact in flow regime than $S_B$. Does this indicate this larger reservoir is contributing to flow regime? Currently there is not adequate analysis to conclude.

reservoir). One reason could be that M.H. Halli sub-basin (with Hemavathi reservoir with a large residence time) receives the highest rainfall compared to other regions in the Upper Cauvery (Reddy et al., 2023) which would have contributed towards a lower impact of $S_B$ compared to $S_A$."

Table 4. The residence time of the reservoirs

| Reservoir | Year of construction | Sub-basin based on gauge location | Catchment area ($10^6$ m$^2$) | Gross Storage ($10^6$ m$^3$) | Residence time (months) |
|---|---|---|---|---|---|
| Harangi | 1982 | Kudige | 419.58 | 240.69 | 7.23 |
| Hemavathi | 1979 | M.H. Halli | 2810 | 1050.63 | 22.63 |
| Krishna Raja Sagara (KRS) | 1938 | Kollegal | 10619 | 1400.31 | 8.68 |
| Kabini | 1974 | T. Narasipur | 2141.90 | 552.74 | 3.57 |

Line 351: Again, it is unclear how the economic value for different crops is estimated, and Section 2.7.1 and equation 2 offers very little description.

A paragraph is added to describe the economic value if crops estimated, the section 2.7.1 is revised as follows:

"The available information on agricultural crops and their distribution is organized at the district level (lowest administrative level within the state boundaries of the states that fall in the basin where such information is available). All the calculations related to these crops are performed at this level, where a total of nine districts are considered in the analysis. By utilizing Geographic Information Systems (GIS), the districts falling within each basin are identified and their areas are determined. Subsequently, using the available data, the area of irrigated and unirrigated land within and outside the basin are calculated. Based on the known cropping patterns for each district, the crops grown are categorized into four growing seasons: kharif, rabi, summer, and annual crops. The area dedicated to each crop is determined proportionally by the acreage of different crops in each district based on the cropping pattern of the respective area.

The maximum yield under irrigated condition and crop prices are obtained from agricultural census sources. Additionally, information on crop coefficients and crop yield response factors is gathered from published literature. An average yearly price is estimated for each crop in all the districts within the studied basin. For irrigated areas, the maximum yield values from the literature are used to calculate the total production. However, for unirrigated areas the reduction in yield is estimated based on the actual evapotranspiration estimates of the hydrological model for each reservoir spatial arrangement. The relationship between crop yield and water depends on the corresponding relative evapotranspiration deficit (potential - actual). The actual yield is calculated based on the following formula by FAO (2012).

$$1 - \frac{Y_a}{Y_o} = K_y \left(1 - \frac{ET_a}{ET_p}\right)$$

Where $Y_a$ = actual Yield, $Y_o$ = optimum Yield, $ET_a$ = Actual Evapotranspiration, $ET_p$ = Potential Evapotranspiration, and $K_y$ = yield response parameter.

Total agricultural production is equal to agricultural output from both rainfed and irrigated areas, with irrigated areas depending on water withdrawn for irrigation. As a result, total agricultural production is a reliable predictor of the amount of water utilized for irrigation. The crop specific prices are multiplied by the corresponding production level to indicate the economic value of the agricultural service supported by the river."

| | |
|---|---|
| Figure 11: Although the mean annual flow is higher in $S_c$ and $S_a$, why is FSR lower in Sc compared to $S_a$ and $S_b$, despite $S_c$ being hydroelectric reservoir? Is it also a function of low flow frequency? I think it require more discussions. And perhaps referring it back to Table 2 would be helpful. | An explanation is provided and included in the revised manuscript as follows:

 "The Fish Species Richness (FSR) is a function of mean annual flow, low flow frequency and other river basin characterstics. Low flow frequency, which refers to the occurrence of periods with minimal water flow, also influences FSR. Fish species adapted to specific flow conditions may be more susceptible to changes in low flow frequency. Reduced low flow periods can result in reduced habitat availability, altered temperature regimes, and increased competition for resources, impacting fish diversity and abundance. Additionally, other river basin |

| | characteristics, such as geomorphology, substrate composition, and habitat complexity, further influence FSR. These factors determine the availability of suitable spawning areas, hiding places, and food resources, thereby shaping the composition and distribution of fish species within the basin (Angermeier & Schlosser, 1989). This study however uses an empirical equation based on various river flow characteristics given the availability of data." |
|---|---|
| Line 395: "The findings show that the scenario without any reservoir ($S_0$) is advantageous for the diversity of fish species." This kind of argument does not offer any practical solution, rather what is the optimum FSR, or what is the critical threshold that should not be exceeded would be helpful. | We acknowledge and agree to your comment and therefore a suitable explanation is provided and it is included in the revised manuscript as follows:

"While this observation does not directly provide a practical solution, it highlights an important consideration for reservoir planning and management. It recognizes the trade-off between the economic benefits of reservoirs for agricultural production and the potential negative impact on fish diversity. However, to address this issue effectively, further investigation and field information are required. To determine an appropriate threshold level a comprehensive assessment of various ecological factors is necessary which includes understanding the specific requirements of fish habitats, their migration patterns, and population dynamics in relation to the presence of reservoirs. Understanding the habitat requirements of different fish species is crucial to identify the potential impacts of reservoirs on their populations. This involves studying factors such as water temperature, dissolved oxygen levels, substrate composition, and availability of food sources. Additionally, assessing the migration patterns of fish can help identify potential barriers created by reservoirs and develop mitigation measures to facilitate their movement. Furthermore, studying population dynamics will provide insights into how the presence of reservoirs affects fish reproduction, growth, and overall population size. All this information is essential for making informed decisions regarding reservoir management practices, such as implementing fish passage systems or establishing protected areas for spawning and nursery habitats." |

| | |
|---|---|
| Section 4.1: As commented above, the discussion seems rather superficial. Looking into seasonal streamflow effects could be an improvement | The discussion is modified and included in the revised manuscript as follows:

"One of the primary effects of reservoirs is flow regulation. By storing water during periods of high flow and releasing it during dry periods, reservoirs can help stabilize streamflow throughout the year. This regulation can reduce the amplitude of seasonal variations, ensuring a more consistent water supply for various sectors such as agriculture, industry, and domestic water use. However, the alteration of natural flow patterns can have ecological implications. Aquatic organisms that rely on specific flow regimes for spawning, migration, or habitat availability may be affected. Disruptions in flow timing and magnitude can impact the reproductive success of fish species and alter the composition of downstream ecosystems.

Reservoirs also play a crucial role in flood control. By storing excess water during periods of heavy rainfall or snowmelt and releasing it gradually, reservoirs can mitigate the risk of downstream flooding. This function is particularly important in regions prone to seasonal floods, protecting communities and infrastructure. However, the alteration of streamflow by reservoirs can have unintended consequences. Changes in flow patterns can affect sediment transport downstream, leading to sedimentation in the reservoir and reduced sediment supply to downstream areas. This can impact river channel stability and downstream ecosystems that rely on sediment inputs. Furthermore, the regulation of streamflow can influence water quality parameters such as temperature, dissolved oxygen, and nutrient levels, with potential impacts on downstream aquatic ecosystems. To mitigate the negative effects and enhance the positive impacts of reservoirs on seasonal streamflow, careful reservoir operation and management are crucial. The inclusion of environmental flows, which aim to maintain a minimum level of water release to sustain downstream ecosystems, is vital to balance human needs with ecological requirements and can ensure the sustainable use of water resources and minimize the impacts of reservoirs on seasonal streamflow dynamics" |

| Technical corrections | Response |
|---|---|
| Just a suggestion, it would be easier for readers to visualize the size and contributing area of the reservoirs if Figure 3 and 4 is combined, and even color code the names of the reservoirs in Figure 3 as in Figure 4. Lat/Long reference in Figure 3 will further improve the figure. | The revised version of Figure 3 includes changes that take into account your comments. Because of technical reasons it was however not possible to merge Figures 3 and 4. |
| Line 182 - 188: It is unclear how the sub-basin is delineated; figure 3 reference would be useful. | The reference to figure 3 is provided. How the watersheds were delineated has been explained above. |
| Table 1: In 1st row, 4th column, what is "For individual reservoir"? | For individual reservoir refers to "combination having one reservoir". It has been made clear in the revised manuscript as "For combination having one reservoir". |
| Line 276: The whole sentence and the meaning of "disentangled" is unclear. | The sentence has been modified and the meaning of disentangled is elaborated in the revised manuscript as follows: "Different combinations of reservoirs are analyzed to understand the flow regimes, utilizing major hydrological indicators like mean annual flow and annual extreme flow conditions. These indicators help in dissecting and understanding the flow patterns. Additionally, the analysis involves classifying the flow regimes based on the storage volume of reservoirs and their spatial configurations" |
| Line 294: "However, in combinations with one and no reservoir..." The sentence is unclear | The sentence has been revised as follows: |

| | "However, in scenarios involving one or two reservoirs despite having varying storage capacities, the extreme low peaks of flows generated by Sa, Sb, Sac, and Sbc appear to be similar" |
|---|---|
| Line 347: Cite the relevant source. | The relevant source is included in the text, as follows:

"Among Horticultural & Plantation crops, the crops like Coffee, Coconut and Cashewnut contribute to 65 percent of the total H&P crops cultivated area (Figure 9, author's estimation).

According to current estimates, the contribution of plantation crops accounts for 58 percent of the economic value of the H&P crops (see Figure 9, author's estimation)'"". |
| Use of sub-basin name and reservoir name (e.g in Figure 8a and b) and throughout the manuscript is inconsistent. Keeping it consistent would help the readers | The figure has been modified along with text in the manuscript where required.

 |

| | |
|---|---|
| Line 363: Generally, instead of saying, "The scenario of 4 dams generates the highest economic value from agricultural production.", it would be better to write, "Presence of all four dams in the basin generates the highest economic value from agricultural production." | The sentence has been modified in the revised manuscript as follows:

"Presence of all four dams in the basin generates the highest economic value from the agricultural production." |
| Figure 9: Please keep the range of vertical axes uniform in all figure panels, if possible. | Due to the variations in the economic value range across different sub-basins, the resulting figure became distorted when using a uniform vertical axis. Therefore, the figure has not been altered and is maintained in its original form. |

**References**

Angermeier, P. L., & Schlosser, I. J. (1989). Species-area relationship for stream fishes. Ecology, 70(5), 1450-1462.

Bekele, E. G., Lant, C. L., Soman, S., & Misgna, G. (2013). The evolution and empirical estimation of ecological-economic production possibilities frontiers. Ecological Economics, 90, 1-9.

Blachly, B., Uchida, E., & Roy, S. G. (2023). Integrating public preferences with biophysical production possibilities: an application to ecosystem services from dam removal. Ecology and Society, 28 (1).

Gómez-Baggethun, E., De Groot, R., Lomas, P. L., & Montes, C. (2010). The history of ecosystem services in economic theory and practice: from early notions to markets and payment schemes. Ecological economics, 69(6), 1209-1218.

Chen, J., Wang, Y., Li, F., & Liu, Z. (2019). Aquatic ecosystem health assessment of a typical sub-basin of the Liao River based on entropy weights and a fuzzy comprehensive evaluation method. Scientific Reports, 9 (1), 14045.

Aazami, J., Esmaili-Sari, A., Abdoli, A., Sohrabi, H., & Van den Brink, P. J. (2015). Monitoring and assessment of water health quality in the Tajan River, Iran using physicochemical, fish and macroinvertebrates indices. Journal of Environmental Health Science and Engineering, 13, 1-12.

Planque, B., Loots, C., Petitgas, P., Lindstrøm, U. L. F., & Vaz, S. (2011). Understanding what controls, the spatial distribution of fish populations using a multi-model approach. Fisheries Oceanography, 20(1), 1-17.

Elsdon, T. S., Wells, B. K., Campana, S. E., Gillanders, B. M., Jones, C. M., Limburg, K. E., ... & Walther, B. D. (2008). Otolith chemistry to describe movements and life-history parameters of fishes: hypotheses, assumptions, limitations and inferences. In Oceanography and marine biology (pp. 303-336). CRC Press.

Bhagat, H., Ghosh, P., & Kumar, D. N. (2021). Estimation of seasonal base flow contribution to a tropical river using stable isotope analysis. Journal of Hydrology, 601, 126661.

Tripti, M., Lambs, L., Gurumurthy, G. P., Moussa, I., Balakrishna, K., & Chadaga, M. D. (2016). Water circulation and governing factors in humid tropical river basins in the central Western Ghats, Karnataka, India. *Rapid Communications in Mass Spectrometry*, *30*(1), 175-190.

Turner, S. W., Bennett, J. C., Robertson, D. E., & Galelli, S. (2017). Complex relationship between seasonal streamflow forecast skill and value in reservoir operations. *Hydrology and Earth System Sciences*, *21*(9), 4841-4859.

Rueda, F., Moreno-Ostos, E., & Armengol, J. (2006). The residence time of river water in reservoirs. Ecological Modelling, 191(2), 260-274.

Reddy, N. M., Saravanan, S., & Abijith, D. (2023). Streamflow simulation using conceptual and neural network models in the Hemavathi sub-watershed, India. *Geosystems and Geoenvironment*, *2*(2), 100153.

Efstratiadis, A., & Koutsoyiannis, D. (2010). One decade of multi-objective calibration approaches in hydrological modelling: a review. *Hydrological Sciences Journal–Journal Des Sciences Hydrologiques*, *55*(1), 58-78.

---

## Author Comment (AC2)

Dear reviewers, thank you for your valuable feedback. We have taken into account the insightful suggestions provided, which have allowed us to elaborate further on the content and improve the overall quality of the paper. The responses to the comments are presented below in a two-column format.

| Comments | Response |
|---|---|
| Comments on how the overall study supports the conclusions presented in the title or discussion/conclusion sections. The title itself suggests that the paper may include a framework that addresses the integrated economic-environmental viability of dams. Indeed, this is a very important issue. However, the paper itself has no actual definition of what a "viable" set of dams is, how the scenarios compared differ in terms of their "viability". The title is therefore misleading. The actual scope is essentially a trade-off analysis between dam implementation scenarios for some of the 4 objectives considered, without offering a critical analysis of the viability of the resulting metrics across scenarios, and simply focusing on comparing values. Is any of the configurations shown in the scenarios economically-environmentally viable, and if so, for whom? In order to coherently approach the statement in the title, I consider that the authors need to provide: A definition of what is a viable configuration of dams. There are several possibilities. For example, it could be based on stakeholder preferences and/or, from a purely phenomenological perspective, the identification of boundaries or tipping points in the system that may preclude the functioning of a key component, e.g., fish biodiversity, agriculture, energy production, etc | The definition of the viability of dams has been provided, along with an explanation to justify it with the present study. The paper has been updated with the following content to address the combined economic-environmental viability of the dams. The following paragraph is added to the revised version of the paper: "A viable configuration of dams refers to a set of parameters and characteristics that consider various factors such as stakeholder preferences and ecosystem preservation to ensure the sustainable and optimal functioning of a dam system. From a stakeholder perspective, it takes into account the preferences and needs of different parties involved, including local communities, government bodies, environmental organizations, and industries. The aim is to strike a balance among diverse interests, incorporating stakeholder preferences into the design and operation of the dam system (Kemmler & Spreng, 2007). From a phenomenological perspective, a viable configuration respects the boundaries within the ecosystem that, if exceeded, could disrupt the functioning of key components such as fish biodiversity, aquatic habitats, and downstream water quality (Kumar and Katoch, 2014). Overall, achieving a sustainable balance between societal needs and environmental protection requires careful planning, scientific analysis, and transparent decision-making processes in dam development (Kemmler & Spreng, 2007; Kumar and Katoch, 2014). In this study, we have chosen agriculture production and fish species richness as indicators to represent both economic development and environmental sustainability. By selecting these components, we aim to find a balance that considers the needs of society while also ensuring the protection of the environment by identifying a specific configuration from a collection of existing dams" |

| | |
|---|---|
| Evidence that the proposed components and indicators are relevant in the context of the proposed case study stakeholders, and that these indicators can be quantified with a reasonable degree of certainty. It is not clear in the paper why the 4 selected indicators were chosen and whether they are representative of system processes or priorities. See comment section 2 for some details. | The selected components and indicators for this study have been justified based on their relevance in assessing the integrated economic and environmental viability of dams. To ensure a comprehensive analysis, two indicators that represent key ecosystem priorities, namely agriculture production and fish species richness, have been chosen. These indicators are considered essential in understanding the impact of the flow regime on the ecosystem. To quantify these indicators with a reasonable degree of certainty, two separate formulas have been employed. |
| | The following paragraph is added to the revised version of the paper: |
| | "Ecosystems have the capacity to provide multiple services simultaneously, but it is generally challenging to manage them in a way that maximizes all services at once (King et al., 2015; Bennett et al., 2009). This can result in trade-offs, where prioritizing one service may come at the expense of others. In river basins, trade-offs often occur as a consequence of management decisions, leading to conflicts between upstream and downstream users. An example is the trade-off between agricultural yield and downstream water quality (Stosch et al.,2019). Another study has examined the cost-effectiveness of hydropower production economics versus salmon habitat restoration costs in relation to the productivity of Atlantic salmon (Bustos et al., 2017). |
| | In the Cauvery basin, approximately 48 percent of the land is used for cultivation (Singh, 2013). In certain stretches of the Cauvery River, there is extensive water abstraction for intensive agriculture (Vedula, 1985; Bhave et al. 2018). This water extraction has resulted in notable changes in the composition of aquatic species, primarily due to the construction of reservoirs. These alterations have had an impact on the overall biodiversity of the river ecosystem. This results in tradeoff between agricultural production and fish species richness. Therefore, these indicators are deemed appropriate for the study." |

| A revised analysis of the interaction between environmental and economic objectives. The current Pareto production frontier analysis only considers the economic component from the perspective of the value of crops, leaving out the monetary value of fisheries, energy, etc. | A partial tradeoff analysis has been conducted, focusing on selected indicators or proxies, while excluding the monetary value of fisheries and energy. This approach allows for a more targeted evaluation of specific factors without incorporating the financial aspects associated with fisheries and energy production. |
|---|---|
| | A comprehensive justification backed up by references has been provided and is included in the paper as follows: |
| | "Cavender-Bares et al. (2015) has demonstrated that for ecological processes and renewable resources, the frontier represents the equilibrium solution derived from a system of equations that represent ecological interactions influencing the yield/synthetic metric/proxy of the two services under consideration. It would also be possible for the axes to represent bundles of services that are assessed using a synthetic metric, indicator, or proxy. The model does not necessitate monetary valuations of ecosystem services. Any quantitative measure of an ecosystem service can be plotted versus another, based on the theoretical or empirically observed relationship governing their joint production (Cavender-Bares et al., 2015). Although the present analysis does not account for the capture of riverine and culture fisheries in reservoirs, it is estimated that the economic value of fisheries is approximately \$0.59 million per year, which is about 12 percent of the economic value of agricultural production (\$5 million per year). Additionally, the analysis does not include electricity generation from only one reservoir as rest of the reservoirs are used for irrigation. Nevertheless, it is recognized that a complete valuation that considers these aspects is imperative, as emphasized in the discussion section." |

Comments on the specific methodologies used to evaluate each of the proposed environmental or economic components of the proposed framework.

In its current form, the paper describes too succinctly many key components of the analytical framework presented in Figure 1, leaving important gaps in the justification of the selected metrics and the methods used to quantify them. It is understandable that, given the conceptual scope of the work, simplification in the main text may be necessary. However, supplementary materials are required to provide additional details that ensure reproducibility and clarity.

Similarly, there appear to be some important limitations in the proposed analytical components. The four most prevalent are as follows:

| Comments | Response |
|---|---|
| • **Hydrological modelling:** It is not clear whether the "landscape hydrological model" is essentially the same as that presented in Ekka et al. (2022), or whether new calibrations were performed for this paper. | The integrated modelling is the same as that presented in Ekka et al. (2022) and that it is used here to simulate runoff at the most downstream gauging stations for various configurations of the reservoirs, where corresponding reservoir models are added or removed from the basin wide FLEX-topo models in a plug and play manner. No new calibration was therefore performed in this paper. |
| • In any case, the reported performance of the model is relatively low (NSE criteria are considered acceptable in the range of +0.2 to +0.5, and good above +0.5. The reported scores are all negative). There is also no mention of the modeling period (assumed to be the same as in Ekka et al., 2022, is 3 years?) Low performance levels can significantly affect the ability to implement data intensive methods such as IHA. | Kindly note that we reported negative (-NSE), since -NSE and MAE were used as objectives of the multi objective optimization algorithm (that finds pareto frontier minimizing these two objectives simultaneously) used to calibrate the integrated model (comprised of Flex-Topo and reservoir model) before its used to simulate stream flows for various reservoir configurations. That means, the negative reverse of NSE value is used to calibrate and validate the model parameters. The -NSE of -0.5 and -0.7 for various calibration and validations steps were reported, which means NSE > 0.5, reasonably good performance. We now explain our response in greater detail and this has been added in the paper as follows: "To calibrate and validate the FLEX-Topo models in Ekka et al. (2022), the dataset of topographic maps, rainfall, and potential evapotranspiration was used. Specifically, the dataset from January 1991 to December 2010 was used for calibration, and the dataset from 2010 to 2016 was used for validation. The reservoir models were calibrated using the dataset composed of inflow, outflow, storage, rainfall, and potential evapotranspiration, for the reservoirs covering the period from January 2011 to December 2016. The Elitist Non-Dominated Sorting Genetic (NSGA-II) algorithm was used to calibrate the model parameters (Deb et al., 2000). Two objective functions are defined and minimized |

simultaneously. The first objective ($f_1$) is the negative of Nash-Sutcliffe Efficiency (NSE) and the second objective ($f_2$) is the Mean Absolute Error (MAE).

$$f_1 = -NSE = -1 + \frac{\sum_{i=1}^{n}(Q_i^m - Q_i^o)^2}{\sum_{i=1}^{n}(Q_i^o - \bar{Q}_o)^2}$$

$$f_2 = \text{MAE} = \frac{1}{n} \sum_{i=1}^{n} |Q_i^o - Q_i^m|$$

Here, $Q_i^m$ is the $i^{th}$ observation for the observed discharge being evaluated. $Q_i^o$ is the $i^{th}$ value of the modelled discharge. $\bar{Q}_o$ is the mean of observed discharge and $n$ being the total number of observations. The parameter sets calibrated for the FLEX-Topo model and the reservoir model are provided in supplementary materials. The NSGA-II parameter setting may have different impacts on computational effectiveness. The results of the calibration and validation after integration of all the reservoirs and measured at the last gauge station is presented in Table 1.

Table 1. The calibration and validation after integration of all the reservoirs

| Model performance | -NSE [range] | MAE [range] (10⁶ m³ day⁻¹) |
|---|---|---|
| Reservoir Calibration (2011-2016) | -0.68 [-0.67 - (-0.69)] | 0.71 [0.70 - 0.72] |
| Flex-Topo Calibration (1991-2010) | -0.53 [-0.54 - (-0.52)] | 0.92 [0.92 -0.97] |
| Flex-Topo Validation (2011-2016) | -0.50 | 0.86 |

| | Within parentheses, the Pareto front ranges produced by the NSGA II algorithm are given for both -NSE and MAE. The MAE is always non-negative, and a lower value means a better prediction. The MAE value was recorded in the range of 0.70 - 0.72 ($10^6$ m$^3$ day$^{-1}$) which is in the acceptable range. Similarly, the -NSE value was observed between -0.51 to -0.73. The -NSE value less than -0.50 is acceptable.

Kindly note our response to the misunderstanding about our reporting of NSE. We had reported negative NSE values instead of positive NSE (due to its use to maximize as an objective in NSGA II). NSE values are greater than 0.5 and sometimes even around 0.7, which indicates reasonably good performance of a model at daily scale that also incorporated reservoir operations. In the current model, the negative reverse of NSE value is used to calibrate and validate the model parameters" |
|---|---|
| IHA methodology application is not sufficiently justifies or documented: In the case of freshwater habitat alterations, the paper selected a subset of IHAs. Why the IHA approach and why a subset of IHA indicators? | The IHA technique is sufficiently elaborated in the revised manuscript as follows: "The Indicators of Hydrological Alteration (IHA), initially proposed by Richter et al. (1996), are used to measure the effects of different reservoir combinations on the flow regime in the Upper Cauvery basin. These indicators consider parameters that have significant relationships with river ecosystems, making them suitable for assessing the impact of dams, barrages, and other water diversion structures on the flow regime. While some other methods of assessing the impact of impoundments on river channels involved field surveys, statistical analyses (Yan, 2010), and geomorphic change detection tools (Wheaton, 2015), the Range of Variability Approach and the associated IHA framework provide a more systematic assessment of flow changes. The IHA method utilizes daily streamflow values and characterizes a flow regime based on factors such as magnitude, duration, frequency, timing, and rate of change of flows. Although the application of the IHA method has been relatively limited in studies of Indian rivers (Mittal et al., 2014, |

| | Kumar and Jayakumar, 2020, Borgohain et al., 2019), this study aims to use the IHA method to gain valuable insights into the impacts of major dams on the flow regime of the Upper Cauvery basin. By doing so, it contributes to a better understanding of the ecological consequences of water diversion and reservoir operations in the region. |
|---|---|
| | Major IHA indicators based on their ecological relevance and their ability to reflect human-induced changes in flow regimes which directly impacts the five groups of hydrological features, that is, flow magnitude, duration, timing, frequency and rate of change are used for the present study". |
| Why not consider other aspects of physical habitat change such as fragmentation, sediment trapping, etc.? | Additional aspects of physical habitat change such as fragmentation, sediment trapping was beyond the scope of the study. However, these additional aspects would be considered in future studies. |
| Also, can the IHAs be calculated with a reasonable level of confidence given the significant margin of error in several flow components of the hydrologic model? | Yes, IHAs can be calculated with a reasonable level of confidence and this has been explained and added in the paper as follows:

"The IHA is typically calculated based on various flow components derived from hydrological models or observed data. While it is true that there can be uncertainties and margin of error associated with flow components in hydrological models, it is possible to calculate the IHA with a reasonable level of confidence, provided that the uncertainties are appropriately addressed. In the present case, the hydrological model is calibrated and validated to ensure that it adequately represents the real-world hydrological processes and assesses its performance." |
| • **Fish species richness is not an ecosystem service.** It is a metric of biodiversity on evolutionary timescales (i.e., how biophysical processes over thousands to millions of years have produced a particular assemblage of species in a region). More importantly, it | Indeed, fish species richness is not an ecosystem service, however, it serves as a measure of biodiversity and play an important role in upholding the river ecosystem health. And therefore, it is crucial to see fisheries from an environmental sustainability point of view. |

| | |
|---|---|
| does not necessarily explain provision services such as fisheries productivity (for example, aquaculture in reservoirs typically has very high productivity with very low biodiversity). | We now explain our response in greater detail and this has been added in the paper as follows:

"Fish species richness refers to the number of distinct fish species found in a specific environment or ecosystem. Although it is not considered a direct ecosystem service, it serves as a measure of biodiversity and reflects the diversity of fish species within a given habitat. Biodiversity, including fish species richness, holds significant importance in ecosystems and possesses inherent value. Fisheries productivity, on the other hand, refers to an ecosystem's capacity to support fish populations that can be harvested for sustenance or other purposes. High species richness can potentially contribute to increased fisheries productivity, as diverse ecosystems often exhibit a range of ecological interactions that sustain robust fish populations.

While the significance of fisheries for human welfare is primarily focused on livelihood generation, food production, and nutritional security, freshwater fisheries offer more than just a source of sustenance and livelihoods (Pownkumar et al., 2022). Fish populations play a vital role in upholding the health of river ecosystems, thereby contributing to the sustainability and resilience of rivers (Holmlund & Hammer, 1999). Unfortunately, the broader role of fisheries is often overlooked in ecosystem management decisions, particularly when freshwater resources are allocated for competing purposes such as irrigation, hydropower, and domestic/industrial use. Therefore, it is essential to see fisheries from an environmental sustainability point of view. While we acknowledge that fish productivity should also be considered alongside agriculture production in economic valuation, it is currently 10 % of the value of agriculture production and therefore is left for consideration in future studies" |
| The estimation of fish species richness was based on a global statistical model developed with the purpose of explaining the global distribution of biodiversity, but NOT of predicting changes in biodiversity based | We agree that the estimation of fish species richness was based on a global statistical model developed by Iwasaki et al. (2012). However, the same formula is validated in 84 major basins |

on short term changes in flows. Also, the cited model was developed based on global datasets with no source data in areas such as the case study, and no reference to validation is made. The proposed model is NOT appropriate for this study, as it suggests that by increasing the mean flows over a few years, you'd

Therefore, the adoption of the Fish Species richness and model based on flow as a predictor of freshwater ecosystem services is not adequate for the analytical purposes stated in the paper and must be revised.

worldwide by Yoshikawa et al. (2014). Therefore, this methodology adopted is adequate for studying the fish species richness in the basin.

We now explain our response in greater detail and this has been added in the paper as follows:

"The FSR (Fish Species Richness) value is derived based on a global statistical model developed by Iwasaki et al. (2012). And the model is being validated in 84 major basins worldwide by Yoshikawa et al. (2014). The value obtained from the equation presented by Iwasaki et al. (2012) is centred in the 20-250 range. Other field studies studies have confirmed that the FSR in cauvery river basin tends to be 146 fish species belonging to 52 families (Koushlesh et al., 2021). In the current study, the estimated FSR for the Cauvery River basin ranges from 70 to 123 species under different projected scenarios. This range of values provides sufficient validation for the results obtained"

Kindly note that the primary objective of using FSR is not to predict FSR values for the basin, but rather to demonstrate how the characteristics of the river basin and its flow can impact fish species richness and different choices of the configuration of the reservoirs can lead to be different economic values and (fish) biodiversity in the long run (since we are using averages of these two variables over 16 years). This, in turn, affects the overall biodiversity of the river ecosystem and subsequently leads to a decline in river sustainability and resilience. By assessing and understanding these relationships, it becomes possible to identify the potential impacts of flow alterations and basin modifications on the long-run biodiversity and ecological stability of the river systems"

| **The Production Possibility Frontier generalizes far beyond the data point ranges.** It is not clear how the authors arrived at the shape of the PPF given the sparse data points of the model output. Also, the | The shape of the PPF was to mimic the convex hull of points in the tradeoff space that correspond to the 16 reservoir configurations.  We acknowledge that the PPF is based on a partial tradeoff |

| | |
|---|---|
| PPF is based on a partial analysis of the monetary value of the system's production and is therefore not representative of the production possibilities of the basin, but only of one sector | analysis between ecosystem services and do not represent the PPF of the whole basin. We give a detailed explanation as follows:

"The shape of the PPF was to mimic the convex hull of points in the tradeoff space that correspond to the 16 reservoir configurations. Since we limited our analysis to the existing set of reservoirs (and did not synthetically include any new reservoirs, which might have provided us with more exhaustive set of points, but this would have been more difficult if not impossible to validate), we also limited our conclusion based on this convex hull, i.e. comparative assessment of dominating and non-dominating reservoir combinations in terms of agricultural production and FSR. To clarify this, we have now updated the figure to clearly show the convex hull.

The central focus of this paper is to evaluate the tradeoff between dominant services that are governed by river flow regime. That is also why the values of fish production and hydropower generation have not been considered in the present analysis. We recognize that constructing a production possibility frontier for all the ecosystem services of the basin is an extensive task that requires substantial data for the analysis which is beyond the scope of this study." |
| Besides the major points mentioned above, it is also worth noting that the formal presentation of contextual data and results, such as maps, tables, and graphs, is sometimes redundant.

For example,

• Table 2 shows the same information as Figures 9 and 10 | Table 2 is deleted.

The figure 2 is moved to the supplementary materials. |

- The maps/graphs shown in Figures 2, 3, 4, and 5 could be combined into a single figure.

However, figure 3 is modified based on query from another reviewer but was impossible to merge with Figure 4. We therefore had to keep both.

[Figure]

Figure 3, An overview of the study area. The reservoirs in the study area are labelled as A , B, C, and D, representing Harangi, Hemavathi, Kabini, and KRS reservoirs, respectively. The labels CA, CB, CC, and CD are used to denote the respective command areas associated with these reservoirs

Figure 5 is conceptual and therefore, it is merged with another figure as indicated below

[Figure]

2 (a) Source : Ekka et al., 2022

2 (b)

| | |
|---|---|
| | 2(a) Modelling concept for individual reservoir: Upstream and downstream contributing areas of the gauging station (GS) are modelled as F1 and F2 respectively. The top row shows how the reservoir model (RM) that contributes to irrigating a certain Command area is integrated with F1 and F2 and calibrated. To simulate the pre-dam situation, RM is removed from the calibrated model, along with its contribution to irrigate the command area

2(b) Integration of reservoir in the basin: All the reservoirs are integrated together to assess the effect of reservoirs on the flow downstream for varying configurations of considered reservoirs. Various configurations of the reservoirs were considered to create a total of 16 different scenarios to assess the potential impacts of flow alterations on the river ecosystem services |
| In conclusion, in the opinion of this reviewer, a major revision of the methods and analysis is required for the publication of the paper. Likewise, supplementary materials with descriptions and data are required to illustrate with sufficient detail each of the analytical components developed. Supplementary data sets to allow for reproducibility are strongly encouraged. | A major revision of the paper has been made in response to the comments received. A supplementary dataset has been included and will be made available along with the manuscript. |

**References**

Barton, D. N., Sundt, H., Bustos, A. A., Fjeldstad, H. P., Hedger, R., Forseth, T., ... & Madsen, A. L. (2020). Multi-criteria decision analysis in Bayesian networks-Diagnosing ecosystem service trade-offs in a hydropower regulated river. Environmental Modelling & Software, 124, 104604.

Blachly, B., Uchida, E., & Roy, S. G. (2023). Integrating public preferences with biophysical production possibilities: an application to ecosystem services from dam removal. *Ecology and Society*, *28*(1).

Bustos, A. A., Hedger, R. D., Fjeldstad, H. P., Alfredsen, K., Sundt, H., & Barton, D. N. (2017). Modeling the effects of alternative mitigation measures on Atlantic salmon production in a regulated river. *Water Resources and Economics*, *17*, 32-41.

Cavender-Bares, J., Polasky, S., King, E., & Balvanera, P. (2015). A sustainability framework for assessing trade-offs in ecosystem services. Ecology and Society, 20(1).

Cooke, S. J., Allison, E. H., Beard, T. D., Arlinghaus, R., Arthington, A. H., Bartley, D. M., et al. (2016). On the sustainability of inland fisheries: Finding a future for the forgotten. Ambio. 45 (7), 753–764. doi:10.1007/s13280-016-0787-4

Ekka, A., Keshav, S., Pande, S., van der Zaag, P., & Jiang, Y. (2022). Dam-induced hydrological alterations in the upper Cauvery River basin, India. *Journal of Hydrology: Regional Studies*, *44*, 101231.

Holmlund, C. M., and Hammer, M. (1999). Ecosystem services generated by fish populations. Ecol. Econ. 29 (2), 253–268. doi:10.1016/s0921-8009(99)00015-4

Iwasaki, Y., Ryo, M., Sui, P., & Yoshimura, C. (2012). Evaluating the relationship between basin-scale fish species richness and ecologically relevant flow characteristics in rivers worldwide. *Freshwater Biology*, *57*(10), 2173-2180.

King, E., Cavender-Bares, J., Balvanera, P., Mwampamba, T. H., & Polasky, S. (2015). Trade-offs in ecosystem services and varying stakeholder preferences: evaluating conflicts, obstacles, and opportunities. Ecology and Society, 20(3).

Koushlesh, S. M. N. S. K., Sajina, A. M., & Roshith, C. M. (2021). Ichthyofaunal diversity of the major Indian rivers: A review. *J. Inland Fish. Soc. India*, *53* (1&2), 22-35.

Pownkumar, V., Ananthan, P. S., Ekka, A., Qureshi, N. W., & Velumani, T. (2022). Fisheries as ecosystem services: A case study of the Cauvery River basin, India. *Front Environ Sci*, *10*.

Stosch, K. C., Quilliam, R. S., Bunnefeld, N., & Oliver, D. M. (2019). Quantifying stakeholder understanding of an ecosystem service trade-off. *Science of the Total Environment*, *651*, 2524-2534.

Yoshikawa, S., Yanagawa, A., Iwasaki, Y., Sui, P., Koirala, S., Hirano, K., ... & Kanae, S. (2014). Illustrating a new global-scale approach to estimating potential reduction in fish species richness due to flow alteration. Hydrology and Earth System Sciences, 18(2), 621-630.